# Dietary L-Tryptophan consumption determines the number of colonic regulatory T cells and susceptibility to colitis via GPR15

Nguyen T. Van[1,2,12], Karen Zhang[1,2,12], Rachel M. Wigmore [1,2], Anne I. Kennedy [1,2], Carolina R. DaSilva [1,2], Jialing Huang [3,11], Manju Ambelil[3], Jose H. Villagomez[1,2], Gerald J. O'Connor [1,2], Randy S. Longman[4], Miao Cao [5], Adam E. Snook [1,2,5], Michael Platten [6,7,8], Gerard Kasenty[9], Luis J. Sigal [1,2], George C. Prendergast [2,10] & Sangwon V. Kim [1,2] ✉

Environmental factors are the major contributor to the onset of immunological disorders such as ulcerative colitis. However, their identities remain unclear. Here, we discover that the amount of consumed L-Tryptophan (L-Trp), a ubiquitous dietary component, determines the transcription level of the colonic T cell homing receptor, GPR15, hence affecting the number of colonic FOXP3$^+$ regulatory T (Treg) cells and local immune homeostasis. Ingested L-Trp is converted by host IDO1/2 enzymes, but not by gut microbiota, to compounds that induce GPR15 transcription preferentially in Treg cells via the aryl hydrocarbon receptor. Consequently, two weeks of dietary L-Trp supplementation nearly double the colonic GPR15$^+$ Treg cells via GPR15-mediated homing and substantially reduce the future risk of colitis. In addition, humans consume 3–4 times less L-Trp per kilogram of body weight and have fewer colonic GPR15$^+$ Treg cells than mice. Thus, we uncover a microbiota-independent mechanism linking dietary L-Trp and colonic Treg cells, that may have therapeutic potential.

Globally, 6.8 million people are affected by inflammatory bowel disease (IBD), a relapsing inflammatory disease of the intestine, and up to 20–50% of patients require surgical resection due to loss of responsiveness to existing medical treatments, posing a serious health risk[1,2]. While IBD is clinically subclassified as Crohn's disease (CD), which affects any part of the gastrointestinal tract, and ulcerative colitis (UC), which is restricted to the large intestine, their causes remain unknown[2,3]. According to genome-wide association studies of IBD patients, over 200 distinct genetic loci have been linked to IBD[4–7]. However, studies of twins have shown that the likelihood of both monozygotic twins developing UC is only up to 17%, compared to 55% in CD, indicating that environmental factors rather than host genetic

[1]Department of Microbiology and Immunology, Sidney Kimmel Medical College, Thomas Jefferson University, Philadelphia, PA, USA. [2]Sidney Kimmel Cancer Center, Jefferson Health, Philadelphia, PA, USA. [3]Department of Pathology, Anatomy, & Cell Biology, Sidney Kimmel Medical College, Thomas Jefferson University, Philadelphia, PA, USA. [4]Jill Roberts Center for IBD, Weill Cornell Medicine, New York, NY, USA. [5]Department of Pharmacology, Physiology, & Cancer Biology, Sidney Kimmel Medical College, Thomas Jefferson University, Philadelphia, PA, USA. [6]CCU Neuroimmunology and Brain Tumor Immunology, German Cancer Research Center, Heidelberg, Germany. [7]Department of Neurology, Medical Faculty Mannheim, MCTN, Heidelberg University, Heidelberg, Germany. [8]DKFZ Hector Cancer Institute at the University Medical Center Mannheim, Mannheim, Germany. [9]Department of Genetics and Development, Irving Medical Center, Columbia University, NY, USA. [10]Lankenau Institute of Medical Research, Wynnewood, PA, USA. [11]Present address: Anatomic Pathology, Geisinger Medical Center, Danville, PA, USA. [12]These authors contributed equally: Nguyen T. Van, Karen Zhang. ✉ e-mail: svkim@jefferson.edu

variants play a dominating role in the onset of UC[8]. Thus, a significant proportion of research efforts to date have been centered on how gut microbiota act on intestinal immune responses, as a major environmental element[9–11]. Even another key environmental component, diet, is mostly known for its indirect impact on intestinal immune responses through altering gut microbial composition and metabolite synthesis[12–17]. However, it is possible that environmental factors independent of the microbiota may substantially influence intestinal immune responses. Identifying these environmental factors and establishing their mechanisms of action are essential if we want to gain a better understanding of the development of inflammatory diseases such as UC.

G-protein coupled receptor 15 (GPR15) is a relatively recent addition to the known cell-surface molecules involved in tissue-specific lymphocyte homing and mediates T cell homing to the large intestine[18]. Recent evidence suggests that aryl hydrocarbon receptor (AhR), which is often regarded as an environmental sensor, is involved in the expression of GPR15[19,20]. While the functions of AhR in the immune system and the maintenance of mucosal surfaces are well studied[21–23], it is rarely known which environmental factors in real-life situations trigger AhR signaling and the expected consequences in human health, due to the unique complexity of AhR-AhR ligand interaction: AhR has a relatively spacious binding pocket for its ligand, permitting the binding and activation of AhR by more than 100 structurally distinct compounds[21,24]. In addition, different AhR ligands can induce distinct patterns of downstream target gene expression in a cell type-specific manner[24–26].

Here, we investigate the effect of L-Tryptophan (L-Trp), a common dietary component, on intestinal T cell responses. We uncover a microbiota-independent mechanism by which L-Trp consumption increases colonic regulatory T (Treg) cells via AhR-GPR15 and reduces the risk of colitis. Our findings suggest that L-Trp could be added to food as a non-invasive therapy to prevent the onset or relapse of UC.

## Results

### L-Trp consumption determines the number of colonic CD4[+] T cells via AhR-GPR15

GPR15 was discovered initially as a co-receptor for HIV and SIV and later identified as a T cell homing receptor for the large intestine[18,27]. GPR15 is expressed preferentially in Treg cells of the large intestine where it is required to guide them for immune homeostasis[18,28,29]. The expression of T cell homing receptors for the skin and the small intestine is controlled by metabolites unique to each tissue, such as vitamin D3 derivatives or vitamin A derivatives, suggesting the role of metabolites in GPR15 expression[30,31]. During a literature search, we encountered data showing that Gpr15 mRNA was among the genes induced in a liver cell line treated with dioxin, a known ligand of AhR[32]. Therefore, to determine whether AhR is required for GPR15 expression in vivo, we used T cell-specific Ahr conditional knockout (CKO) mice (Ahr-CD4 CKO) and their littermates. At steady state, Ahr-deficiency in T cells significantly reduced Gpr15-expressing CD4[+] T cells in the large intestine lamina propria (LILP) regardless of their Foxp3 expression (Supplementary Fig. 1a). Interestingly, GPR15 expression in CD8β[+] T cells and CD4[-]CD8β[-] double-negative (DN) T cells in the LILP was not reduced in the absence of AhR (Supplementary Fig. 1b), indicating that AhR-mediated GPR15 regulation is restricted to CD4[+] T cells. During the preparation of this manuscript, two additional studies supported our findings in both mice and humans[19,20]. Since AhR acts upstream of GPR15, we sought to identify the environmental factors that may induce GPR15 via AhR, the molecular mechanisms of GPR15 induction, and the consequences of AhR activation by these environmental factors on gut health (Fig. 1a).

In exploring relevant environmental factors affecting GPR15 expression and immune homeostasis in the colon, we focused initially on L-Trp, which can be converted to numerous AhR ligands in vivo[33].

L-Trp is an essential amino acid that must be obtained from the diet or microbiota, but it appears that the contribution of de-novo synthesized L-Trp by gut microbiota is minimal[34]. Hence, we hypothesized that the amount of ingested L-Trp could influence the availability of specific AhR ligands responsible for GPR15 expression and impact T cell responses in the large intestine. To test this hypothesis, we have prepared three different protein-free, isocaloric, elementary diets containing specific amounts of each amino acid: Trp-C (normal range of L-Trp level in conventional mouse diet), Trp-Sup (L-Trp supplementation), and Trp-Def (no L-Trp) (Supplementary Table 1). Dietary L-Trp supplementation significantly increased GPR15[+]CD4[+] T cells in the LILP after three weeks, but not GPR15[-]CD4[+] T cells (Fig. 1b, c). Ahr-CD4 CKO mice exhibited no increase in response to L-Trp supplementation (Fig. 1c), indicating that L-Trp-mediated increase of GPR15[+] T cells is dependent on AhR in T cells. Notably, this increase occurred only in the LILP, but not in the spleen (Fig. 1d). Consistent with the findings with Trp-Sup, the Trp-Def diet decreased GPR15[+]CD4[+] T cells in the LILP (Fig. 1c). However, Trp-Def also reduced both GPR15[+] and GPR15[-] cells in the spleen regardless of AhR (Fig. 1d), indicating AhR- and GPR15-independent phenotype predominates in the absence of L-Trp in the spleen.

We determined the minimum duration of dietary L-Trp supplementation required to increase GPR15[+]CD4[+] T cells in the LILP. We observed that two weeks of 7-fold L-Trp supplementation was sufficient to result in a significant rise in GPR15[+]FOXP3[+]CD4[+] cells in the LILP at a steady-state (Fig. 1e). We reasoned that the increase of GPR15[+] cells occurred specifically in the LILP after dietary L-Trp supplementation because increased GPR15 expression can enhance GPR15-mediated homing of T cells to the LILP as previously demonstrated[18]. To test this hypothesis, we fed Gpr15[gfp/gfp]Foxp3[mrfp] mice with elementary diets and determined the fate of GPR15-wannabe (GFP[+]) cells in the absence of GPR15. Without GPR15, GFP[+] cells do not increase in the LILP upon L-Trp supplementation (Supplementary Fig. 2). Instead, GFP[+] cells slightly increased in the spleen (Supplementary Fig. 2). These results indicate that dietary L-Trp supplementation increases GPR15[+] CD4[+] T cells in the LILP via an AhR-mediated increase in GPR15[+]CD4[+] T cells and their subsequent migration to the LILP via GPR15.

CD4[+] T cells exhibited dominant expression of GPR15 in the LILP and CCR9 in the small intestine lamina propria (SILP) (Fig. 2a), suggesting a compartmentalized T cell homing strategy between the small and large intestines. Since the CCR9-inducing retinoic acids were unable to induce GPR15 expression[18], we investigated if L-Trp supplementation could increase CCR9 expression in vivo. L-Trp supplementation did not affect the expression of CCR9[+] or FOXP3[+] cells in the LILP and SILP (Fig. 2b, Supplementary Fig. 3a). L-Trp treatment increased GPR15[+] Treg cells in the mesenteric lymph nodes (MLNs) draining the proximal and middle colon (Supplementary Fig. 3b), demonstrating that L-Trp-mediated induction of GPR15 may occur in the draining MLNs. We also found that most of the colonic GPR15[+] Treg cells that increased after L-Trp supplementation were HELIOS[+] Treg cells, as opposed to microbiota-derived RORγt[+] Treg cells (Fig. 2c)[35–38].

### L-Trp enhances GPR15 via AhR during CD4[+] T cell activation and migration to the large intestine

Our studies depicted above were performed at a steady state without regard to the history of the GPR15[+]CD4[+] T cells with elevated GPR15 expression. To determine if dietary L-Trp supplementation enhances GPR15 expression during initial T cell activation by physiological antigens and their subsequent migration to the LILP and whether L-Trp supplementation increases peripherally-derived Treg cells, we prepared HH7-2 T cell-receptor (TCR) transgenic mice (HH7-2[Tg]Rag1[n/n]) with different congenic markers (Cd45[(1/1)] or Cd45[(1/2)]) in the presence (Ahr[fl/fl]) or absence (Cd4[cre]Ahr[fl/fl]) of AhR in T cells. It has been shown that HH7-2[+] naive CD4[+] T cells migrate to the LILP upon recognition of

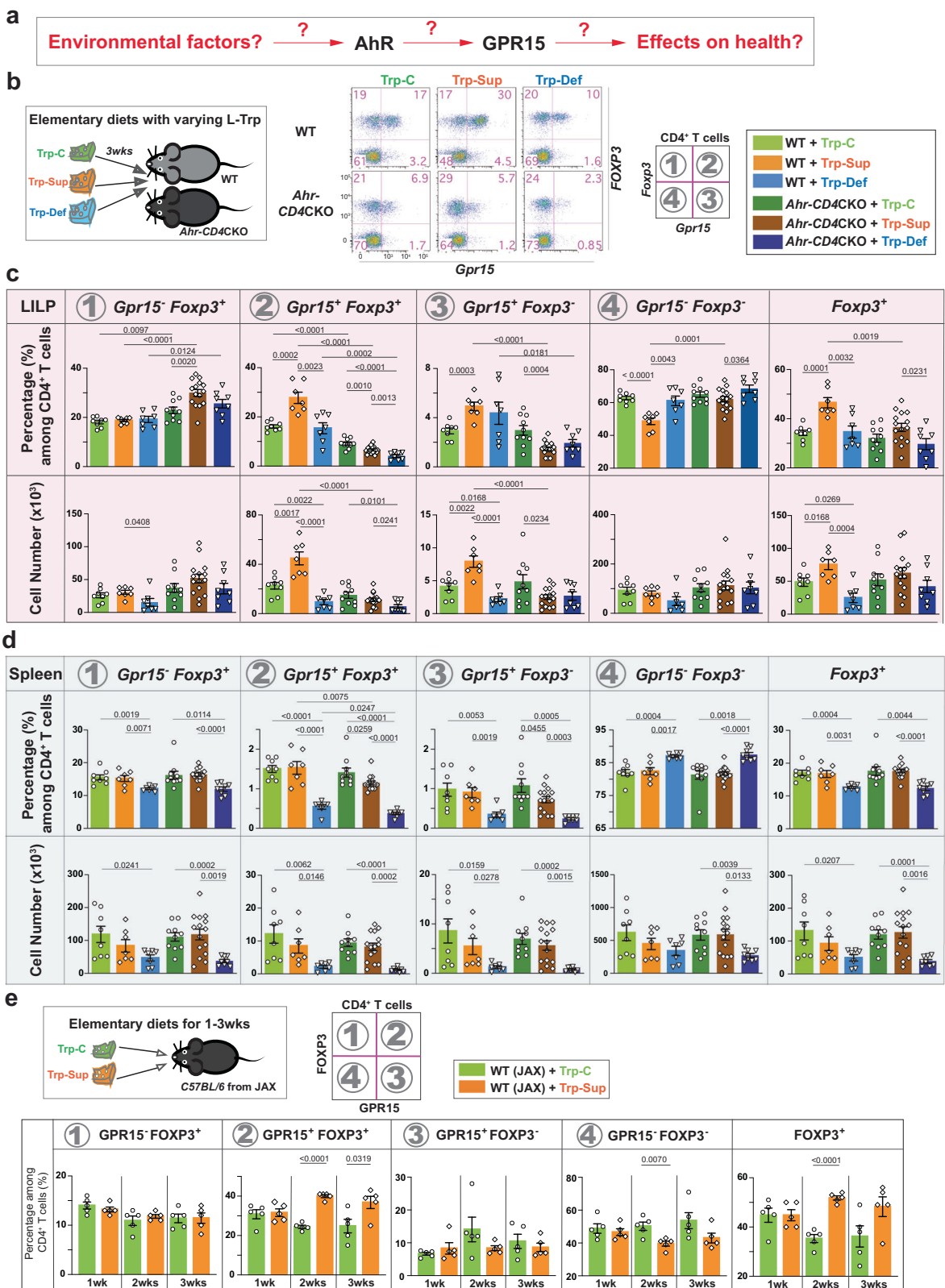

microbial antigens[39]. Recipient mice ($Cd45^{(2/2)}$) were colonized with *Helicobacter hepaticus* (*H. hepaticus*) carrying antigens recognized by HH7-2 TCRs. Subsequently, Trp-C or Trp-Sup elementary diets were fed to recipient mice, and 3 days later, a 1:1 mixture of CD4[+] HH7-2[+] naive T cells from WT and *Ahr-CD4* CKO mice was transferred (Fig. 2d). Recipient mice were analyzed on day 10 after the donor-cell transfer, at which point we started to observe a significant number of donor T cells

in the LILP. Only WT GPR15[+]FOXP3[+] donor cells increased upon L-Trp supplementation, and this phenotype was found specifically in the LILP but not in the SILP or spleen, consistent with steady-state data at least for GPR15[+]FOXP3[+] Treg cells (Fig. 2e, f). These results indicate that dietary L-Trp can increase GPR15 expression via AhR during naive T cell differentiation to peripherally-induced Treg cells and their subsequent migration to the LILP.

**Fig. 1 | The amount of ingested dietary L-Trp determines the number of colonic CD4⁺ T cells via the AhR-GPR15 pathway. a** Major questions addressed in our study. **b** Three different types of elementary diets (Trp-C (green); Trp-Sup (orange); Trp-Def (blue), Table S1) were fed to mice with TJU microbiota 1 for three weeks. Representative flow cytometry plots for CD4⁺ T cells in the LILP were shown. **c**, **d** *Gpr15*-GFP and *Foxp3*-mRFP expression was examined in CD4⁺ T cells in the LILP (**c** in the pink shade) and the spleen (**d** in the light blue shade) from wild-type (WT: *Ahr^(fl/fl)Gpr15^(gfp/+)Foxp3^mrfp*) and *Ahr-CD4*CKO mice (*Cd4^CreAhr^(fl/fl)Gpr15^(gfp/+)Foxp3^mrfp*). 8–14-week-old mice were used. Data are presented as mean values ± SEM of the percentage of each population of CD4⁺ T cells or the cell numbers. The numbers of mice used (*n*) for WT group were 8 for Trp-C, 7 for Trp-Sup, and 7 for Trp-Def. The mice used for *Ahr-CD4*CKO group were 10 for Trp-C, 15 for Trp-Sup, 8 for Trp-Def. Representative of three independent experiments. **e** 8–10-week-old WT mice from the Jackson Laboratory (JAX) were treated with Trp-C or Trp-Sup elementary diets for 1–3 weeks (1wk (*n* = 5), 2wk (*n* = 5), 3wk (*n* = 5)). Data are presented as mean values ± SEM of the percentage of each population of CD4⁺ T cells. Representative of two independent experiments. Each data point represents the result from one mouse, and *p* values were calculated by two-sided student's t-test (**c**–**e**). Source data are provided as a Source Data file (**c**–**e**).

## L-Trp increases GPR15⁺ Treg cells via host IDO1/2 enzymes

Ingested L-Trp is known to be converted by microbiota to various metabolites that affect gut health (Fig. 3a)[40,41]. Therefore, we analyzed whether different microbiota affect the phenotype following L-Trp supplementation. Increased GPR15⁺FOXP3⁺ Treg cells in the LILP in response to dietary L-Trp supplementation was observed in four representative conditions with diverse microbiota (Thomas Jefferson University [TJU] microbiota 1 and 2, Jackson laboratory [JAX] microbiota, and Taconic Biosciences [TAC] microbiota), indicating that differences in the microbiota of experimental mice do not affect this phenotype (Figs. 1c, e, 2b, 3b, c). However, increased GPR15⁺FOXP3⁻ T effectors in the LILP upon L-Trp supplementation were observed only in mice with a certain TJU microbiota (#1), but not in mice with another TJU microbiota (#2), JAX microbiota, or TAC microbiota (Figs. 1c, e, 2b, 3b, c). GPR15⁺FOXP3⁺ Treg cells were not reduced in germ-free (GF) mice compared to specific pathogen-free (SPF) mice at steady state, whereas GPR15⁺FOXP3⁻ T effectors significantly decreased in GF mice without changes in GPR15⁻FOXP3⁻ T effectors (Fig. 3d). Mice lacking microbiota after treatment of 4 different antibiotics still responded to L-Trp supplementation and exhibited the increase of GPR15⁺FOXP3⁺ Treg cells, but not GPR15⁺FOXP3⁻ T effectors (Fig. 3e). These findings demonstrated that dietary L-Trp supplementation increases GPR15⁺FOXP3⁺ Treg cells regardless of microbiota, while GPR15⁺FOXP3⁻ T effectors require a specific microbiota to respond to L-Trp supplementation, suggesting that the microbe-derived Trp metabolites are not required for GPR15 expression in FOXP3⁺ Treg cells but may be required in FOXP3⁻ T effectors.

Two different pathways can metabolize L-Trp in the human body: the kynurenine pathway in which initial metabolism is mediated by three different enzymes producing kynurenine (TDO, IDO1, IDO2), or the serotonin pathway in which L-Trp is converted to 5-hydroxy tryptophan (5-HTP) by two different enzymes (TPH1, TPH2) (Fig. 3a)[40,42]. To determine which host pathways of L-Trp metabolism are involved in GPR15 induction in FOXP3⁺ Treg cells, we examined GPR15 and FOXP3 expression in CD4⁺ T cells in the LILP of mice deficient in these enzymes at steady state. IDO1 and IDO2 are primarily involved in immunological functions, while TDO is mainly involved in liver metabolism[42,43]. We found that GPR15⁺FOXP3⁺ Treg cells in the LILP decreased significantly in *Ido2* single KO mice and even more so in *Ido1/2* double KO (DKO) mice (Fig. 3f). In contrast, TDO deficiency did not change GPR15 expression in CD4⁺ T cells in the LILP (Supplementary Fig. 4a). While the decrease of GPR15⁺ Treg cells in *Ido1/2* DKO mice was partial (Fig. 3f), this partial decrease is likely due to the compensation by their common downstream metabolite, produced by TDO in *Ido1/2* DKO mice. We believe that preferential expression of IDO1 and IDO2 by immune cells may affect the local availability of metabolites in the draining lymph nodes or the LILP even in the presence of L-Trp metabolites produced in other tissues. In the serotonin pathway, TPH2 is specifically expressed in the brain, so we tested the effect of TPH1[44]. We observed that *Tph1* deficiency did not change GPR15 expression in CD4⁺ T cells in the LILP (Supplementary Fig. 4b). These results indicate that the microbiota-independent, host L-Trp metabolism by IDO1/2 is a default pathway in response to L-Trp, selectively increasing colonic Treg cells via AhR and GPR15. In contrast, the L-Trp-mediated increase of colonic FOXP3⁻ T effectors occurs only in the presence of specific microbiota.

## AhR ligands determine Treg or T effector cell selective-GPR15 expression

The observation that L-Trp supplementation promotes GPR15⁺ Treg cells, but not GPR15⁺ T effectors, was puzzling. Since it is known that different AhR ligands trigger distinct sets of target gene expression in a cell-context-dependent manner[24,26] and that different AhR ligands can induce either Treg cells or Th17 cells via AhR[25], we hypothesized that the identity of AhR ligands enriched in the body after L-Trp consumption might determine GPR15 expression in either Treg or T effector cells. To test this, 62 compounds with known or potential AhR ligand activity based on a PubMed search were examined (Supplementary Table 2). We found that only 7 of 62 compounds were capable of inducing *Gpr15*-GFP expression in CD4⁺ T cells in vitro via AhR (Fig. 4a and Supplementary Fig. 5). This indicates that not all AhR ligands are the same and can induce GPR15 expression in CD4⁺ T cells. Using antibody staining, the activity of these seven compounds was further confirmed in both mouse and human T cells (Fig. 4b).

Among microbe-derived compounds, we found that indole-3-pyruvic acid (I3-PYA), indoxyl sulfate (I3-S, produced by the conversion of microbe-derived indoles in the liver), and pyocyanin induced the GPR15 expression above the background level in an AhR-dependent manner (Fig. 4b and Supplementary Fig. 5). Our detection of pyocyanin is consistent with recently published data regarding pyocyanin activity in human T cells[20]. None of the known plant-derived AhR ligands, including indole-derivatives such as indole-3-carbinol (I3-CBL), could directly induce GPR15 expression in vitro (Fig. 4a). Among host-derived metabolites tested, only 6-formylindolo[3,2-b]carbazole (FICZ) appeared to induce GPR15 expression[45]. While none of the known host metabolites in the kynurenine pathway induced GPR15 expression in vitro, data from *Ido1/2* DKO mice (Fig. 3f) suggest that uncharacterized Trp metabolites downstream of IDO1/2 are likely to exist and can induce GPR15 expression in Treg cells. Notably, several xenobiotic compounds exhibited GPR15-inducing activity via AhR, including ANI-7, benzo[a]pyrene (BaP), and beta-naphthoflavone (β-NP) (Fig. 4a, b and Supplementary Fig. 5)[45–48].

We found that FICZ and BaP can induce GPR15 expression preferentially in either FOXP3⁻ T effector or FOXP3⁺ Treg cells, respectively (Fig. 4b). This result confirms our hypothesis that the identity of AhR ligands can determine GPR15 expression in either T effector or Treg cells. The identification of BaP was especially intriguing since BaP is enriched in cigarette smoke[49]. Multiple studies have found that GPR15 in blood T cells is one of the most prominent markers for cigarette smokers[50–52]. Therefore, BaP in cigarette smoke may induce GPR15 in T cells via AhR in humans. To determine whether BaP can increase GPR15⁺FOXP3⁺ Treg cells in vivo in the LILP, HH7-2 TCR⁺CD4⁺ naive T cells were transferred to *H. hepaticus*-colonized recipient mice, which were then injected intraperitoneally with varying concentrations of BaP (Fig. 4c). We discovered that BaP could significantly and dose-dependently increase colonic GPR15⁺FOXP3⁺ Treg cells (Fig. 4c). Therefore, our data indicate that different AhR ligands determine T cell type-specific GPR15 expression.

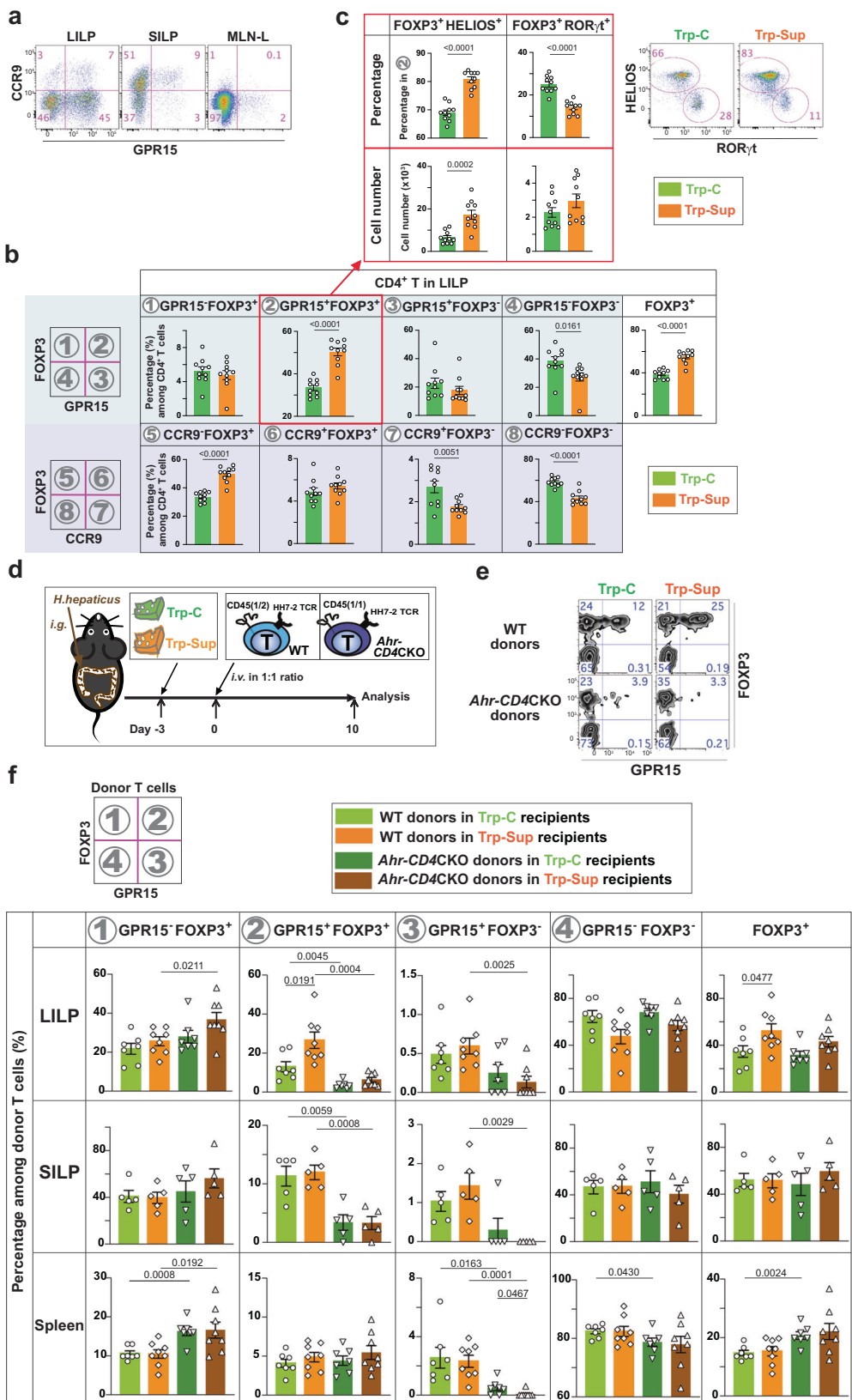

## AhR directly controls the transcriptional activation of *Gpr15*

While we and others[19,20] demonstrated that AhR functions upstream and regulates GPR15 expression (Supplementary Fig. 1a-b), the molecular mechanisms of how AhR induces GPR15 expression is not clear. To test the role of AhR in *Gpr15* transcription, we used *Gpr15^gfp* allele in *Ahr-CD4* CKO. We found that the absence of AhR reduces the GFP

reporter expression (Fig. 5a), indicating that the transcription of *Gpr15* gene is controlled by AhR. At a genomic locus located approximately 10 kb downstream of the 3' end of the GPR15-coding region[53], we identified two potential binding sites conserved between mouse and human for AhR, two for AP-1, one for STAT, and one for FOXO1 (Fig. 5b). Recent CHIP data suggest that this region likely contains

**Fig. 2 | L-Trp supplementation increases GPR15 expression via AhR during CD4⁺ T cell activation and their migration to the large intestine. a** Representative flow cytometry plot of CD4⁺ T cells in the LILP, SILP, and mesenteric lymph nodes-draining the proximal and mid-colon (MLN-L) in wild-type (WT) mice from JAX. **b** CD4⁺ T cells in the LILP, SILP, and MLN-L of WT mice from JAX with 8–12 weeks of age were analyzed for GPR15 (in the light blue shade) or CCR9 expression (in the light purple shade) in addition to FOXP3 expression after elementary diet treatment. The number of mice used: 10 for Trp-C and 10 for Trp-Sup. **c** GPR15⁺ Treg cells in the LILP in **b** were analyzed for HELIOS and RORγt expression. Trp-C (*n* = 10), Trp-Sup (*n* = 10). Combined results of two independent experiments and representative flow cytometry plots (**b, c**). **d–f** WT mice (JAX) were gavaged with *H. hepaticus* and treated with Trp-C or Trp-Sup elementary diets. After 3 days of elementary diets,

recipient mice were transferred intravenously with 1:1 mixture of 10⁴ naive CD4⁺ T cells from two different mice (WT: *HH7-2^Tg Rag1^(n/n) Ahr^(fl/fl) CD45^(1/2)*; Ahr-CD4CKO: *HH7-2^Tg Rag1^(n/n) CD4^cre Ahr^(fl/fl) CD45^(1/1)*) and analyzed 10 days later. **d** Schematic plan. **e** Representative flow cytometry plots of donor cells in the LILP. **f** The percentages of different populations among total donor T cells are shown. Data are presented as mean values ± SEM. The number of mice used: *n* = 7 for LILP, *n* = 5 for SILP, *n* = 7 for spleen for recipients treated with Trp-C; *n* = 8 for LILP, *n* = 5 for SILP, *n* = 8 for spleen for recipients treated with Trp-Sup. Representative of two independent experiments. 8–12-week-old mice were used (**a–c, e, f**). Data are presented as mean values ± SEM (**b, c, f**). Each data point represents the result from one mouse, and *p* values were calculated by two-sided student's t-test (**b, c, f**). Source data are provided as a Source Data file (**b, c, f**).

functional binding sites for AhR in vivo[19]. To determine whether AhR binding to these two tandem sequences of dioxin-responsive element (DRE) is required for GPR15 expression, we generated two mouse strains: mice with an allele called *Gpr15^TFΔ*, in which the entire conserved region containing multiple transcription factor (TF) binding sites (the two AhR binding sites, two AP-1 binding sites, and a STAT binding site) was deleted; and mice with an allele called *Gpr15^DREmut1* in which both AhR binding sites were mutated to prevent AhR binding (Fig. 5b). We crossed the mice heterozygous for those alleles with *Gpr15^(gfp/gfp)* mice, allowing us to determine cis-regulatory effects of TF-binding sites on GPR15 protein expression in the absence of another copy of the *Gpr15* gene in trans. GPR15⁺CD4⁺ T cells in the LILP were severely reduced to a similar degree in both *Gpr15^TFΔ* and *Gpr15^DREmut1*, indicating that AhR binding to the two tandem binding sites is required to properly express GPR15 (Fig. 5c). However, deficiency of AhR in T cells and the mutations in AhR binding sites did not eliminate GPR15 expression in CD4⁺ T cells (Fig. 5c, Supplementary Fig. 1b), indicating that additional pathways may be capable of inducing GPR15 expression, such as through STAT3 and TGFβ, as shown previously[18]. Deleting the entire conserved region in *Gpr15^TFΔ* allele decreased GPR15 expression also in CD8⁺ and DN T cells in the LILP, indicating that TF-binding sites other than AhR-binding sites affect GPR15 expression in these cells (Supplementary Fig. 6).

### L-Trp supplementation reduces the future risk of developing colitis through GPR15

As a default pathway, the L-Trp-mediated increase in GPR15⁺FOXP3⁺ Treg cells occurs consistently regardless of gut microbiota of the host. In contrast, L-Trp-mediated increase of GPR15⁺ T effectors occurs only in the presence of particular microbiota and this increase is marginal (Figs. 1c and 3a, b). Typically, the significant increase or decrease of Treg cells in the LILP is accompanied by a corresponding decrease or increase of local inflammatory T cells[18]. Some GPR15⁺CD4⁺ T cells are poised to express effector cytokines such as IFNγ or IL-17A, but it is GPR15⁻CD4⁺ T cells that represent the majority of T helper cells in the LILP at steady state (Supplementary Fig. 7a). We found that L-Trp supplementation could reduce IFNγ- and IFNγ-/IL-17A-producing CD4⁺ T cells, and IL-17A-producing γδ T cells in the LILP (Supplementary Fig. 7b, c). Therefore, we hypothesized that dietary L-Trp supplementation can reduce the risk of colonic inflammation by increasing GPR15⁺ Treg cells via the AhR-GPR15 pathway. To test this, we utilized a mouse colitis model generated by *Citrobacter rodentium* (*C. rodentium*) which mimics human colitis induced by attaching and effacing enterotoxigenic *E. coli*[54] and another colitis model induced by Dextran Sulfate Sodium (DSS). The literature demonstrated that the abundance of GPR15⁺ Treg cells in the LILP affects the severity of *C. rodentium*-mediated colitis[18] and Treg cells are important for suppressing acute colitis induced by DSS[55], confirming that these models are appropriate for testing the effect of the increased GPR15⁺ Treg cells. To minimize the effect of extra L-Trp on the growth of *C. rodentium* and pathology, we treated the mice with either control diet or diet supplemented with L-Trp for two weeks prior to the colitis induction and then switched to

control diet (Fig. 6a). We observed that prior L-Trp supplementation significantly reduced the inflammation and pathology of colitis in WT mice, but not in *Gpr15* KO mice both in *C. rodentium*-mediated colitis (Fig. 6b, c, and) and DSS colitis (Fig. 6d, e). These results indicate that dietary L-Trp supplementation can prevent the future risk of developing colitis by increasing GPR15⁺ Treg cells in the LILP[18]. Additionally, we determined if L-Trp supplementation could be used to treat colitis as well as prevent it. Five days after *C. rodentium* gavage, we began feeding mice the Trp-Sup diet, at which point the mice developed colitis signs such as diarrhea. In this situation, we discovered that Trp-Sup did not ameliorate the symptoms of *C. rodentium*-induced colitis (Supplementary Fig. 8).

Our findings highlight that L-Trp food supplementation, in addition to microbiota manipulation as suggested by others[10,56], may serve as a non-invasive, preventative therapy for the onset or relapse of UC. However, while human Treg cells do express GPR15[28,29,52], overall frequency of GPR15⁺ Treg cells in the human colon is less than in the mouse colon (Fig. 6f, g)[18,28]. Based on the literature[57] and after accounting for the fact that the metabolic rate per kilogram of body weight in mice is seven times that of humans[58] (see the calculation in Methods), we found that mice consume three to four times more L-Trp than humans (Fig. 6f). This difference may explain why mouse large intestines contain more GPR15⁺ Treg cells than human large intestines (Fig. 6g)[18,28]. It also suggests that humans may consume more L-Trp to increase colonic GPR15⁺ Treg cells for therapeutic purposes. Single doses of 100 mg of L-Trp per kg of body weight is known to have no adverse effects[59].

### Discussion

In this study, we discovered a connection between dietary L-Trp and colonic T cell responses. The amount of L-Trp consumed determines the transcription level of the colon T cell homing receptor, GPR15, by AhR and, consequently, the number of CD4⁺ T cells specifically in the colon via GPR15-mediated homing (Figs. 1, 5, and Supplementary Fig. 2). Intriguingly, our data indicate that colonic GPR15⁺ Treg cells development is independent of microbiota, whereas GPR15⁺FOXP3⁻ T cell development is dependent on microbiota (Fig. 3d). At steady state in our experimental conditions with four distinct microbiota, dietary L-Trp supplementation primarily increases HELIOS⁺ Treg cells which can be thymus-derived although an alternative origin exists[60], with marginal or no effect on GPR15⁺FOXP3⁻ T cells in the large intestine (Figs. 1e, 2b, c, 3b, c). These results prompted us to pursue two separate lines of inquiry: first, what are the effects of L-Trp supplementation in disease conditions? Second, why does L-Trp supplementation increase GPR15⁺ Treg cells preferentially?

First, we discovered that only two weeks of dietary L-Trp supplementation was sufficient to increase the number of colonic GPR15⁺ Treg cells and reduce the future risk of experimental colitis (Fig. 6), which demonstrates how influential dietary L-Trp is on colonic T cell responses. Considering the ubiquitous nature of L-Trp in all protein-based diets, our results suggest that continuous consumption of L-Trp^lo or L-Trp^hi diets or the total amount of L-Trp consumed over a

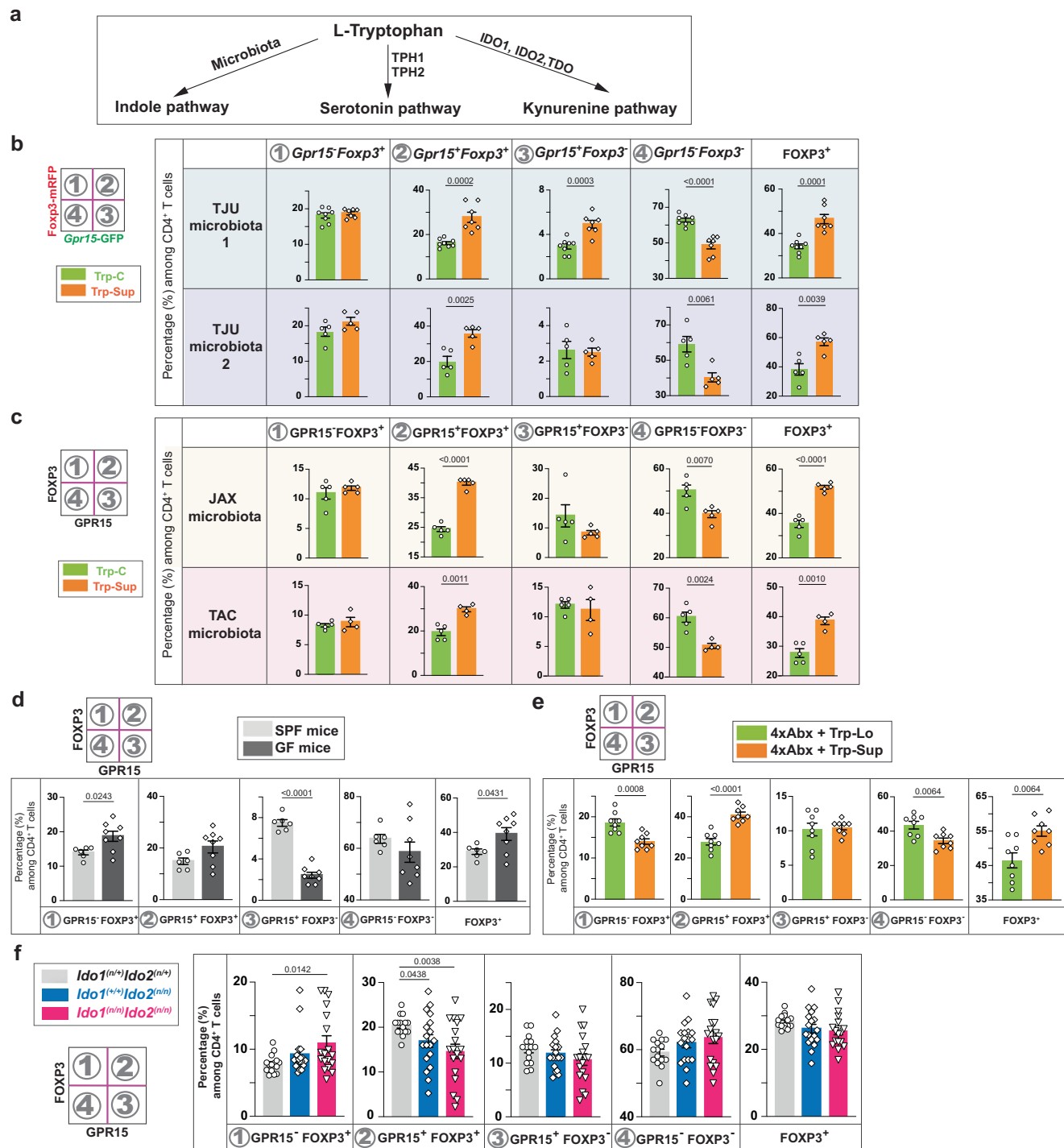

**Fig. 3 | The default response to L-Trp is a selective increase of GPR15⁺ Treg cells via host IDO1/2 enzymes in a microbiota-independent manner. a.** L-Trp metabolic pathways upon ingestion. **b-f.** Percentages of each population among CD4⁺ T cells in the LILP based on *Gpr15*-GFP and *Foxp3*-mRFP expression (**b**) or GPR15 and FOXP3 expression (**c–f**). **b, c** Wild-type (WT) mice (*Gpr15*(gfp/+)*Foxp3*(mrfp)) with Thomas Jefferson University (TJU) microbiota (two different microbiota: TJU1 or TJU2) (**b**), or WT mice with Jackson laboratory microbiota (JAX) or Taconic Biosciences microbiota (TAC) (**c**) were treated with Trp-C or Trp-Sup elementary diets for 2 wks. The number of mice used (*n*): 8 for TJU1 (Trp-C), 7 for TJU1 (Trp-Sup), 5 each for TJU2 (Trp-C and Trp-Sup), 5 each for JAX (Trp-C and Trp-Sup), 5 for TAC (Trp-C), 4 for TAC (Trp-Sup). **d** Specific pathogen-free (SPF) mice (TAC, *n* = 6) and germ-free

mice (*n* = 8) were compared at a steady state. **e** WT mice with JAX microbiota were treated with four different antibiotics (4xAbx) in drinking water for four weeks. Elementary diets (Trp-Lo (*n* = 8) or Trp-Sup (*n* = 8)) were provided for two weeks in the presence of 4xAbx. Representatives of two independent experiments (**b**, **c**, **e**). **f** *Ido1/2*(n/+), *Ido2*(n/n), and *Ido1/2*(n/n) mice were analyzed at a steady state. Combined results of five independent experiments. The number of mice used: 15 for *Ido1/2*(n/+), 18 for *Ido2*(n/n), 18 for *Ido1/2*(n/n). 8–14-week-old mice were used (**b-f**). Data are presented as mean values ± SEM (**b–f**). Each data point represents the result from one mouse, and *p* values were calculated by two-sided student's t-test (**b–f**). Source data are provided as a Source Data file (**b–f**).

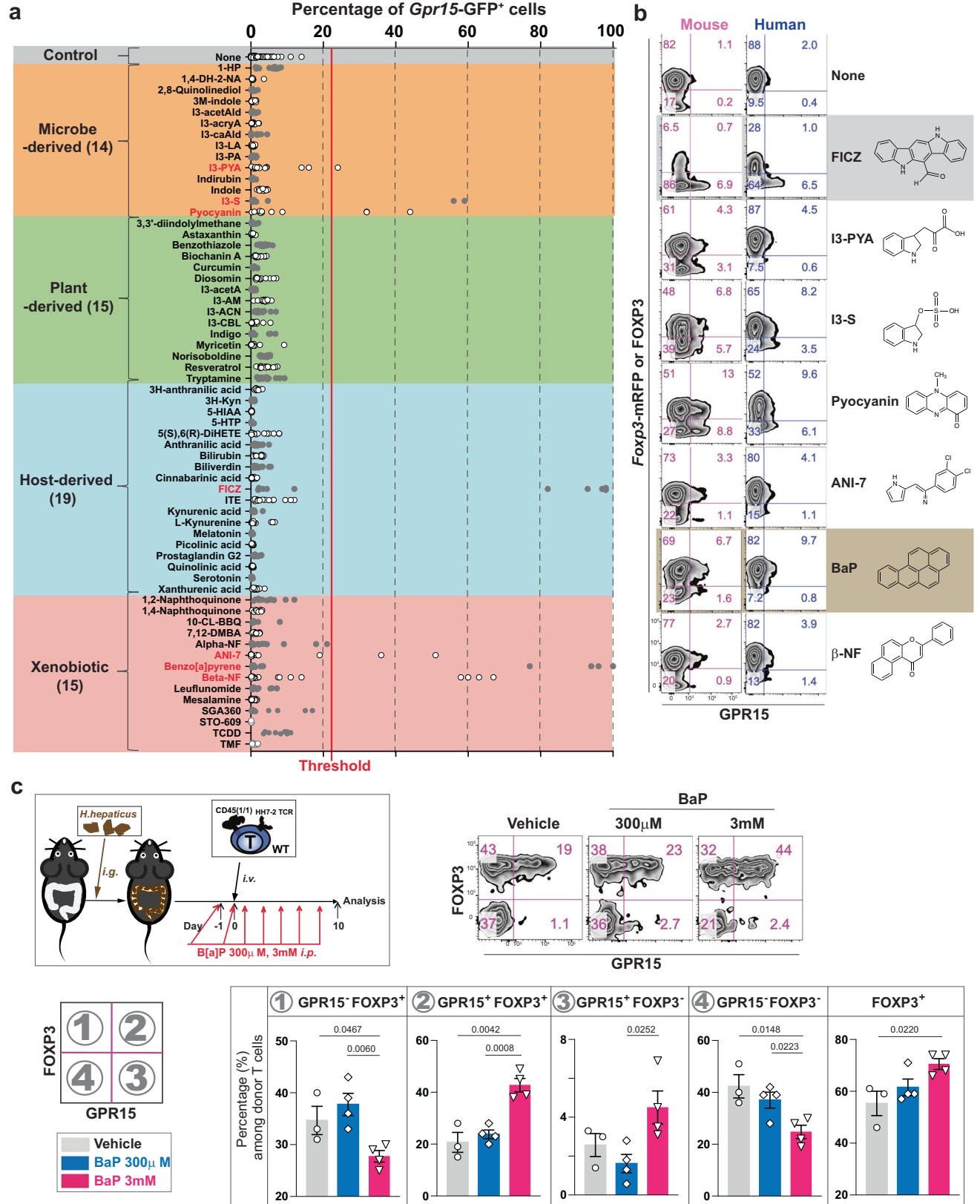

certain period may have a substantial effect on colonic immune homeostasis. This also suggests the intriguing concept of increasing dietary L-Trp consumption as a non-invasive therapeutic strategy to enhance colonic GPR15⁺ Treg cells and prevent the onset or relapse of ulcerative colitis. In support of this, we found that humans consume three to four times less L-Trp than mice, which explains why the overall

frequency of GPR15⁺ Treg cells in the human colon is lower than in the mouse colon (Fig. 6f, g)[18,28]. While L-Trp supplementation before colitis induction alleviated symptoms (Fig. 6a–e), L-Trp supplementation after colitis induction did not (Supplementary Fig. 8). These results suggest a potential limitation of L-Trp-mediated therapies. It is well established, however, that none of the single mouse colitis models

**Fig. 4 | The identity of AhR ligand determines the T cell types expressing GPR15.** **a** Chemical compounds with known AhR ligand activity (Table S2) were categorized based on their origin and tested on CD4[+] naive T cells from *Gpr15(gfp/+)Foxp3mrfp* mice. The red vertical line is set as a threshold for further analysis. Seven compounds (labeled red) induced a GFP signal above the threshold. **b** Compounds shown to induce GPR15 expression in Fig. 4a were tested with mouse and human CD4[+] T cells for GPR15 and FOXP3 protein expression. Representative flow cytometry plots of at least three independent experiments. **c** Wild-type mice (JAX) were gavaged and injected intraperitoneally with 300 μl of either PBS (*n* = 3) or 300 μM (*n* = 4) or 3 mM (*n* = 4) of BaP 7 times. GPR15 and FOXP3 expression of T cells with HH7-2 TCRs were analyzed on day 10. The percentage of each population in CD4[+] donor T cells in the LILP is shown. Representative of two independent experiments. Data are presented as mean values±SEM (**c**). Each data point represents the result from one mouse, and *p* values were calculated by two-sided student's t-test (**c**). 8-14-week-old mice were used (**a**–**c**). Source data are provided as a Source Data file (**a**, **c**).

replicate all the characteristics of human IBD[61]. In the future, it may be necessary to assess the efficacy of L-Trp supplementation in a more clinical setting. In addition, there is evidence that GPR15 recruits Treg cells into colorectal cancer tissues, thereby influencing anti-tumoral immunity[62], suggesting a potential side effect in cases of colorectal cancer.

Regarding the second question, we discovered that GPR15[+] Treg cell development requires L-Trp metabolism by host IDO1/2 enzymes (Fig. 3e). But none of the known IDO1/2 downstream metabolites induce GPR15 expression (Fig. 4a) and we were unable to identify the host-derived AhR ligands for GPR15[+] Treg cell development in vivo. However, our comprehensive in vitro screening of 62 compounds with AhR ligand activity showed that not all AhR ligands are the same: only 7 out of 62 compounds directly induce GPR15 expression (Fig. 4a). In addition, some AhR ligands can enrich either GPR15[+]FOXP3[+] Treg cells (BaP) or GPR15[+]FOXP3[-]CD4[+] T effectors (FICZ), validating our hypothesis that the identity of AhR ligands determines which T cell type expresses GPR15 (Fig. 4b). Thus, the overall outcome of L-Trp consumption is likely determined by which AhR ligands are produced in greater quantities. In this scenario, dietary L-Trp supplementation may produce more AhR ligands through IDO1/2, which can promote GPR15 expression in Treg cells. This idea of dominant metabolite production by IDO1/2 upon L-Trp ingestion is supported by the previous research indicating that about 95 % of L-Trp absorbed by the host is utilized by the host kynurenine pathway, in which IDO1/2 play a role, whereas only 4–6 % and 1–2 % of L-Trp ingested are utilized by gut microbiota and the serotonin pathway, respectively[63,64]. Still, it is possible that FICZ or microbiota-dependent metabolites such as I3-PYA, I3-S, and pyocyanin play a role in the production of GPR15[+]FOXP3[-] T cells in vivo (Fig. 4a, b). GPR15[+]FOXP3[-] T cell level varies a lot in human probably due to the diversity of human gut microbiota (Fig. 6g). Further investigation into the identities of AhR ligands and gut microbes that trigger GPR15[+]FOXP3[-] T cells could shed light on this issue.

The identification of BaP as the AhR ligand increasing the ratio of GPR15[+] Treg cells to T-effectors (Fig. 6b) is particularly interesting. BaP is generated by the incomplete combustion of organic molecules and can be found in high concentrations in coal tar, cigarette smoke, and charred meat[45,49]. Well-established clinical observations in patients with inflammatory bowel disease suggest that cigarette smoking reduces the onset of UC while exacerbating CD[65]. Given that UC occurs exclusively in the large intestine, and that BaP most strongly increases the GPR15[+] Treg-to-T-effector ratio (Fig. 4b), cigarette smoking may prevent the onset of UC by increasing GPR15[+] Treg cells via BaP. Further studies are required to understand the exact mechanisms underlying the protective effects of cigarette smoking on UC.

Our study uncovered a mechanism by which L-Trp in the diet regulates colonic Treg cells via the IDO1/2-AhR-GPR15 pathway, resulting in a sustained effect on immunological homeostasis in the colon. The protective effects of L-Trp or its metabolites on different models of experimental colitis were shown previously[66–69]. However, those effects are likely mediated by mechanisms distinct from ones we characterized here, such as gut epithelial barrier maintenance or microbiota changes[67,70–72], because the L-Trp metabolites used in some of the previous studies[67,68] do not induce GPR15 expression (Fig. 4a) and a sustained effect of the L-Trp supplementation was not tested in

those previous studies. While L-Trp supplementation predominantly increased HELIOS[+] Treg cells in the majority of our experimental settings (Fig. 2c), it can also increase peripherally-derived Treg cells (Fig. 2f). Consequently, it remains unknown whether AhR ligands downstream of IDO1/2 induce GPR15 only in already differentiated Treg cells or affect Treg cell differentiation, particularly in relation to multiple pathways for the development of peripherally-derived Treg cells[73]. In addition, AhR can mediate the increase in colonic Treg cells via a mechanism distinct from what we discovered here[74] and AhR is implicated in mucosal immune responses mediated by cell types other than Treg cells[75–79]. Therefore, further research is necessary to evaluate the effect of L-Trp supplementation on these cell types, the sources of AhR ligands for each cell type, and their respective contributions to inflammatory responses during colitis.

## Methods

### Ethics

All animal studies were performed according to the protocol approved by the Institutional Animal Care and Usage Committee (IACUC) of Thomas Jefferson University. For human specimens, normal human intestinal endoscopic biopsy samples were obtained under Weill Cornell Medicine IRB approved protocol (1103011578) or normal adjacent colon tissues were obtained from patients undergoing colorectal cancer resection under Thomas Jefferson University IRB protocol 18D.495. Before being received for analysis, all human specimens were de-identified. As a result, we lack information on sex-, gender, race, ethnicity, or other socially relevant groupings.

### Mice

All mice used were on C57BL/6 background. Wild-type C57BL/6 mice were obtained from either Jackson Laboratory or Taconic Biosciences. *Gpr15(gfp/gfp)* mice were described previously[18]. *Ahr (fl/fl)* mice, *CD4cre* mice, *Cd45(1/1)* mice, and *Thy1(1/1)* mice were obtained from Jackson Laboratory[80,81]. *Foxp3mrfp* mice, *Ido2(n/n)*, *Ido1/2 (n/n)*, *Tdo2 (n/n)*, *Tph1(n/n)*, and HH7-2 TCR transgenic mice were described previously[39,43,44,82–84]. *Ahr(n/n)* mice were generated by crossing *Ahr (fl/fl)* mice with *EIIacre* mice from Jackson Laboratory[85]. *Gpr15TFΔ*, *Gpr15DREmut1* mice were generated by improved-Genome editing via Oviductal Nucleic Acids Delivery (iGONAD) method at Thomas Jefferson University[86]. Briefly, guide RNA complex was composed of tracrRNA (Integrated DNA Technologies) and the following crRNA (Integrated DNA Technologies) targeting sequences: 5′ -TGCCAGTTAAAA-CAGTtcTG-3′, 5′ -CTATTTCCACCCAAGAagAC-3′. Single-stranded repair DNA template was prepared by PCR of the genetic locus with 5′-CCA AGT TTA TCT CAG AAG GAA GAG GAA ACA GA-3′ and 5′-ATC TAT TTG AGA GGA AGT AAA CCA CAA GGG GA-3′, followed by treatment of PCR products with T7 exonuclease (New England Biolab). A mixture of CAS9 protein (PNA Bio), gRNA complexes, and repair template was injected into the oviduct of plugged female mice and electroporation was done through the oviduct[86]. Germ-free mice were provided by a Germ-free facility at New York University. For the analysis, mice were euthanized by $CO_2$ gas or by exsanguination/perfusion according to the IACUC-approved animal protocol. Sex was not considered in this study design since our preliminary experiments with diets of various L-Trp contents showed no differences based on sex.

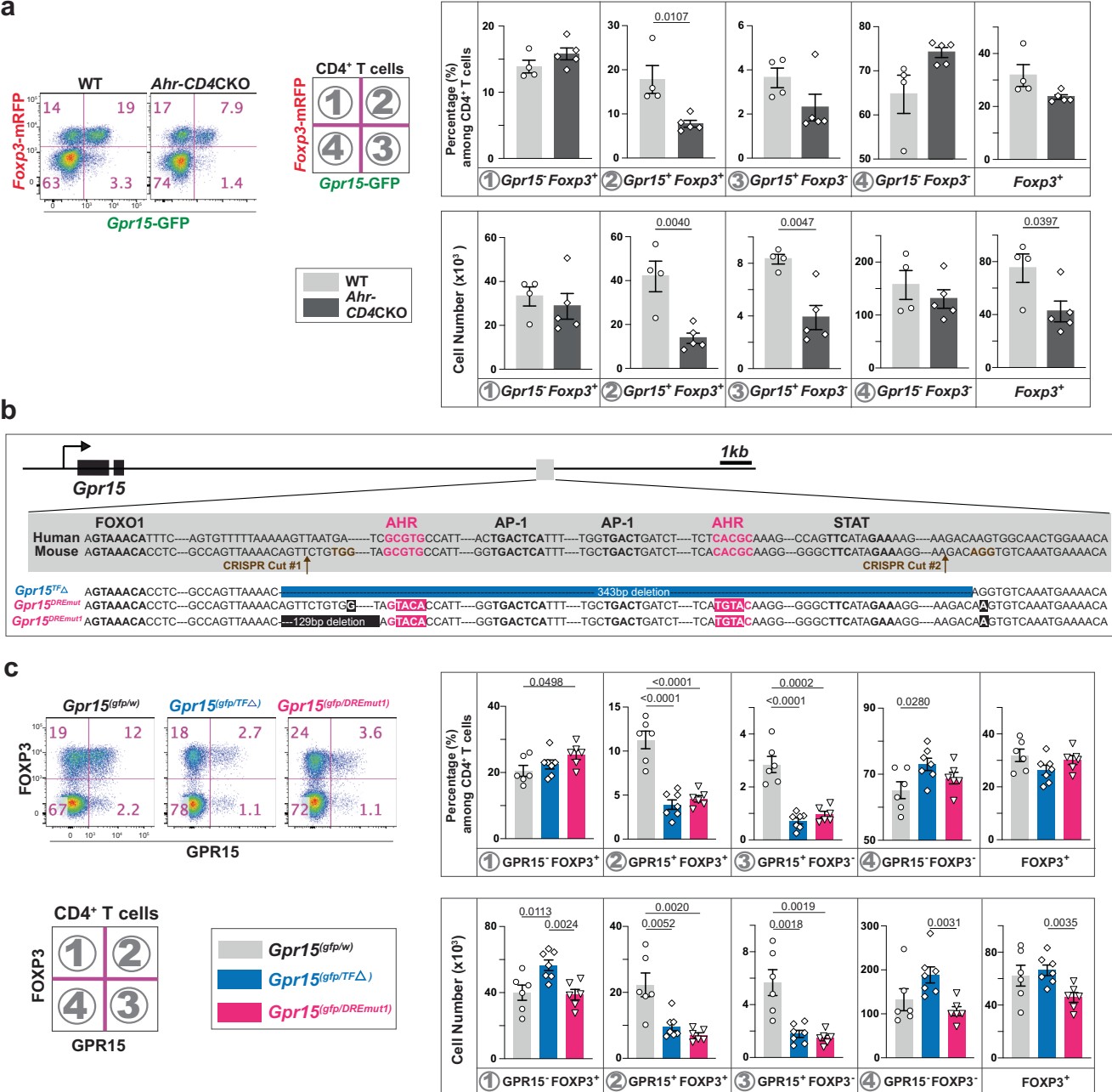

**Fig. 5 | AhR directly controls the transcriptional activation of *Gpr15* in CD4⁺ T cells via dioxin-responsive elements (DRE). a** *Gpr15* and *Foxp3* mRNA expression in CD4⁺ T cells in the LILP was determined by the reporter expression in the presence (WT: *Ahr*⁽ᶠˡ/ᶠˡ⁾*Gpr15*⁽ᵍᶠᵖ/⁺⁾*Foxp3*ᵐʳᶠᵖ, *n* = 4) or the absence (*Ahr-CD4*CKO: *CD4*ᶜʳᵉ*Ahr*⁽ᶠˡ/ᶠˡ⁾*Gpr15*⁽ᵍᶠᵖ/⁺⁾*Foxp3*ᵐʳᶠᵖ, *n* = 5) of AhR. Flow cytometry plots and the percentage/cell number of each population among CD4⁺ T cells in the LILP are shown. Representatives of at least two independent experiments. **b** The DNA sequence of the conserved region in the 3′-distal to *Gpr15* locus. Our knock-in strategy is shown: *Gpr15*ᵀᶠᐞ allele has a deletion of 343 bps containing two AhR binding sites, two AP-1

binding sites, and a STAT binding site. Also, we intended to mutate two AhR binding sites to make a *Gpr15*ᴰᴿᴱᵐᵘᵗ allele but ended up with a *Gpr15*ᴰᴿᴱᵐᵘᵗ¹ allele having an additional 129 bp deletion of the non-conserved sequence. **c** Representative flow cytometry plots and the percentage/cell number of two independent experiments are shown for CD4⁺ T cells in the LILP. The number of mice used: 6 for WT, 7 for *Gpr15*ᵀᶠᐞ, and 6 for a *Gpr15*ᴰᴿᴱᵐᵘᵗ¹. 8–14-week-old mice were used (**a**, **c**). Data are presented as mean values±SEM (**a**, **c**). Each data point represents the result from one mouse, and p values were calculated by two-sided student's t-test (**a**, **c**). Source data are provided as a Source Data file (**a**, **c**).

## Special diets for mice

Mice in the TJU animal facility were fed normally with LabDiet 5010. To control the amount of L-Trp ingestion by mice, we obtained elementary diet from Envigo, in which the known quantity of amino acids without protein was incorporated during the manufacturing of the chow pellets. Four different types of elementary diets (Envigo) were used (Table S1): Trp-C (Envigo, TD.07788) has 1.8 g/kg of L-Trp. Trp-Sup (Envigo, TD. 170745) has 12.5 g/kg of L-Trp. Trp-Lo

(TD.200535) has 0.18 g/kg of L-Trp. Trp-Def (Envigo, TD.08467) has no L-Trp. The total nitrogen intake of mice remained the same across four different elementary diets by adjusting the amount of non-essential amino acids. For L-Trp supplementation before *C. rodentium*-induced colitis (Fig. 6b, c), we used modified diet 5010Trp (Envigo, TD.210218) in which 15.15 g/kg of L-Trp was added to ground LabDiet 5010. As a control to "5010Trp" modified diet, we used LabDiet 5010 that went through the same process of grinding and

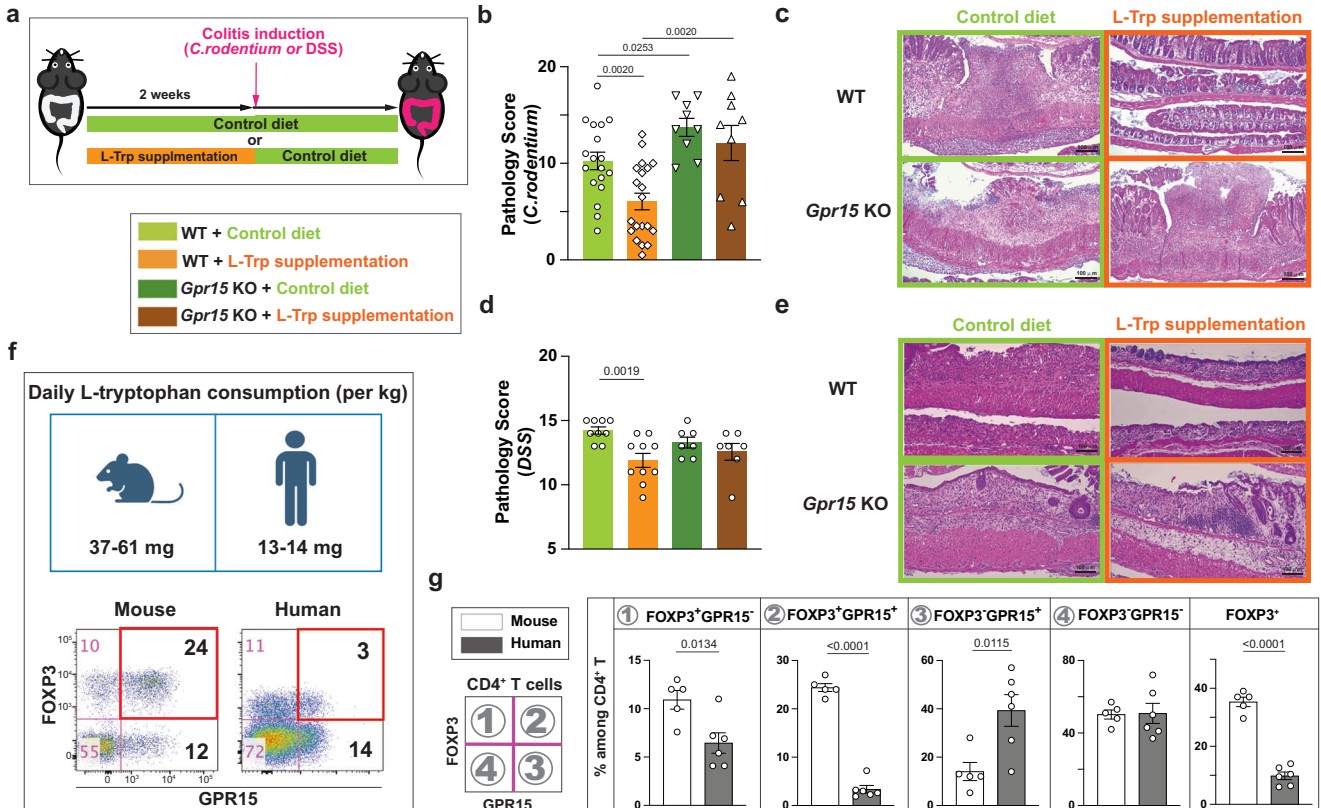

**Fig. 6 | L-Trp supplementation reduces the future risk of developing colitis through GPR15.** **a**–**e** Wild-type (WT) mice or *Gpr15*^(gfp/gfp) (*Gpr15* KO) mice with JAX microbiota were fed with control diet or L-Trp supplemented diet for two weeks. Both mice were switched to control diet and gavaged with *C. rodentium* or treated with DSS. Mice were analyzed at day 10 (for *C. rodentium*) or day 7 after colitis induction (for DSS). **b** Pathology score of the large intestine with *C. rodentium*-induced colitis. Number of mice used: 18 for WT with Control diet, 20 for WT with L-Trp supplementation, 9 for *Gpr15* KO with Control diet, and 9 for *Gpr15* KO with L-Trp supplementation. Combined results of four independent experiments. **c** Representative images of large-intestine tissues (*C. rodentium*-induced colitis), stained with Hematoxylin and Eosin (H/E). The bars represent 100 μm. **d** Pathology

score of the large intestine with DSS-induced colitis. Number of mice used: 9 for WT with Control diet, 10 for WT with L-Trp supplementation, 7 for *Gpr15* KO with Control diet, and 7 for *Gpr15* KO with L-Trp supplementation. Combined results of two independent experiments. **e** Representative images of large-intestine tissues (DSS-induced colitis). The bars represent 100 μm. **f, g** Estimated daily L-Tryptophan consumption between mice and humans after normalization and the expression of GPR15 and FOXP3 among CD4+ T cells in the large intestine are shown (*n* = 5 for mice, *n* = 6 for humans). 8–14-week-old mice were used (**b**–**g**). Data are presented as mean values±SEM (**b**, **d**, **g**). Each data point represents the result from one mouse or one human, and p values were calculated by two-sided student's t-test (**b**, **d**, **g**). Source data are provided as a Source Data file (**b**, **d**, **f**, **g**).

repacking to pellets as "5010 C" without adding L-Tryptophan (Envigo, TD.160112). The normal level of L-Trp in the mouse diet (TrpC or 5010) in our experiments ranges from 1.8–3 g of L-Trp/kg of diet. Since we observed that 10 adult mice split between two cages consumed about 500 g of chow pellets in 14 days, consistent with the literature[87], we estimated that one mouse (on average 25 g of body weight) consumes about 6.4–10.7 mg of Trp /day, which equals to 256–428 mg of L-Trp/kg of body weight/day. In comparison to that, it has been estimated that the daily intake of L-Trp in adult humans is estimated to be between 900–1000 mg/day[57]. With adult human average weight of 70 kg, L-Trp intake in humans ranges between 12.8–14.3 mg L-Trp/kg of body weight/day. Since mice require to have seven times more metabolic rate than humans[58], daily intake of L-Trp/kg of body weight of mice were normalized by dividing it by 7 (37–61 mg /kg of body weight) for the comparison with humans (Fig. 6f).

### Antibiotic treatment of mice
For antibiotic treatment, a mixture of ampicillin (1 g/L), vancomycin (0.5 g/L), metronidazole (1 g/L), neomycin (1 g/L), and Kool-Aid (Grape flavor, 20 g/L) was filtered through 0.22 μm and provided in the drinking water.

### Cell preparation from the intestines
For the large intestine, cecum and colon were collected, cut open, and cut into 4 pieces without cecal patches. They were washed with ice-cold PBS by hand-shaking and treated with 1 mM DTT/PBS for 10 min, followed by two consecutive treatments with 30 mM EDTA/PBS at room temperature in the shaker (100 rpm/min). Tissue pieces were washed with PBS once, cut into smaller pieces of about 0.5 mm×0.5 mm, and incubated in the digestion solution (collagenase 8 [Sigma], dispase [Worthington], DNase I [Sigma] in RPMI with 10% fetal bovine serum) at 37 °C with occasional shaking for 1–1.5 h. Digested tissues were filtered through 100 micron filter and cells were recovered at the interface between 40% and 80% percoll (GE healthcare) after spinning at 850 g for 20 min. For the small intestine, Peyer's patches were removed, cut open, and cut into 8–12 pieces. Subsequently, samples were washed with ice-cold PBS once and treated with 5 mM EDTA/PBS for 10 min at the shaker (100 rpm/min) set at 37 °C. The rest of the steps for cell isolation after EDTA/PBS treatment is the same as described above for the large intestine. Human samples were processed in the same procedure as described for the mouse large intestine except that collagenase D (Roche) was used instead of collagenase 8 (Sigma) and that dispase (Worthington) was not used in the digestion solution. The following antibodies were used for the staining:

**Table 1 | Criteria for evaluating pathology (Fig. 6)**

| Criteria\Score | 0 | 1 | 2 | 3 |
|---|---|---|---|---|
| Epithelial regeneration | Complete | Slight injury | Surface not intact | No tissue repair |
| Goblet cell depletion | >50/HPF | 25–50/HPF | 10–25/HPF | <10/HPF |
| Epithelial hyperplasia | None | 1–50% | 51–100% | >100% |
| Atrophy/Crypt loss | Normal crypt | Mild | Moderate | Severe |
| Crypt damage | Intact | Basal 1/3 | Basal 2/3 | Entire loss |
| Ulceration/erosion | None | Focal, lamina propria | Muscularis propria | Full thickness |
| Edema | None | Mild | Moderate | Severe |

APC-hGPR15 (Clone: 367902, R&D systems or Clone: SA0302A10, Biolegend), eFluor 450-hFOXP3 (Clone: 236 A/E7, Thermofisher), PECy7-CD3ε (Clone: UCHT1), APC780-CD4 (Clone: RPA-T4).

### Staining of GPR15, FOXP3, and cytokines in lymphocytes

Anti-mouse GPR15 (S15042I, BioLegend) was biotinylated and used for the staining. For transcription factor staining, the FOXP3/Transcription factor buffer set (eBiosciences) and the following antibodies were used: FOXP3 (Clone: FJK-16s, Thermofisher), RORgt (Clone: B2D, Thermofisher), and HELIOS (Clone: 22F6, Biolegend). For intracellular cytokine staining, single-cell suspension prepared from the large intestine was stimulated in the presence of Monensin (BD GolgiStop) with PMA (50 ng/ml) and Ionomycin (500 ng/ml) for 3 hrs. Cells were additionally stained by fixable dye for dead cells (Invitrogen) before going through BD Fix/Perm kits (BD) for intracellular cytokine staining with the following antibodies: IFN-γ (Clone: XMG1.2, Biolegend), IL-17A (Clone: eBio17B7, Thermofisher), and IL-4 (Clone:11B11, Biolegend).

### *Helicobater hepaticus* colonization and T cell transfer

*Helicobacter hepaticus* (*H. hepaticus*) was inoculated in Blood agar plate (TSA with 5% Sheep Blood, Fisher Scientific) and cultured for 2–3 days in the anaerobic chamber with 80% $N_2$, 10% $CO_2$, and 10% $H_2$. Cells were harvested by scraping the surface of the culture plates. OD 600 = 1 was $1 \times 10^8$ CFU/ml. Each mouse was gavaged with $2–4 \times 10^7$ CFU, and their colonization was confirmed by PCR of their fecal DNA with the following sequences: 5′ -ATGGGTAAGAAAATAGCAAAAA-GATTGCAA-3′ and 5′ -CTATTTCATATCCATAAGCTCTTGAGAATC-3′. HH7-2 TCR expressing naive CD4$^+$ T cells were isolated from spleen and lymph nodes of HH7-2 TCR transgenic mice in *Rag1*$^{(n/n)}$ background. Subsequently, single-cell suspension was treated with AKC lysis buffer and enriched with the MojoSort mouse CD4 T cell isolation kit (Biolegend). A total of $1 \times 10^4$ CD4$^+$ naive T cells were transferred to each recipient mouse colonized with *H. hepaticus*.

### Citrobacter rodentium-induced colitis

*Citrobacter rodentium* strain DBS100 (American Type Culture Collection 51459) was inoculated in 2 ml LB broth and cultured at 37 °C overnight. The next day, the bacteria was inoculated in fresh medium (500 ml) and grown until the culture reached the exponential phase (O.D.$_{600}$ = 0.4–0.6). O.D.$_{600}$ = 1 was $2.5 \times 10^8$ bacteria. Mice starved for 12 h prior were gavaged with $2 \times 10^{10}$ bacteria and provided with food after that. For the prevention model of colitis (Fig. 6b, c), colonic tissues were analyzed at day 10 after the gavage by H&E staining. For the treatment models of colitis (Supplementary Fig. 8), 5010 C and 5010Trp modified diets were provided on day 5 after the gavage of *C. rodentium*. Mice were analyzed 7 days later.

### DSS-induced colitis

Dextran sulfate sodium (#DB001, TdB labs) was used to induce colitis. Mice with JAX microbiota were fed with Trp-C or Trp-Sup elementary diet for two weeks and were switched to the normal LabDiet 5010 at the beginning of DSS treatment (3% DSS in drinking water with 20 g/ liter of grape-flavored Kool-Aid). On day 5 of DSS treatment, DSS water was removed, and normal water was provided. Mice were analyzed on day 7 (Fig. 6d, e).

### H&E staining, histopathological analysis

Representative sections with Hematoxylin/Eosin staining were evaluated in a double-blind manner by two gastrointestinal pathologists for the following criteria in the Table 1. In brief, the assessment of goblet cell loss and mucosal neutrophil infiltration was done microscopically at a magnification of 40x, while the remaining histologic parameters were evaluated at 10x. Representative histologic changes were photographed (Fig. 6c, e). Pathology score was calculated by combining the scores from readings on epithelial regeneration, goblet cell depletion, epithelial hyperplasia, atrophy/crypt loss, crypt damage, ulceration/erosion, and edema (Table 1) as described previously[18].

### Screening of AhR ligands for GPR15 induction in vitro

All T cell culture was done in RPMI supplemented with 10% Fetal Bovine Serum (Gemini), 10 mM of HEPES, 10 mM of Sodium-Pyruvate, 10 mM of Non-Essential Amino Acids, and 0.055 mM of β-mercaptoethanol. For mouse T cells, naive T cells (CD4$^+$CD62L$^{hi}$CD44$^{lo}$CD25$^-$mRFP$^-$GFP$^-$ or CD4$^+$CD62L$^{hi}$CD44$^{lo}$CD25$^-$GPR15$^-$) were sorted from *Gpr15*$^{gfp/+}$*Foxp3*$^{mrfp}$ mice (BD Aria II sorter) and stimulated with anti-CD3ε (Clone: 145-2C11, BioLegend), anti-CD28 (Clone: 37.51, BioLegend), and plate-bound anti-hamster rabbit-IgG antibodies (MPbio) in the presence of human TGFβ (Peprotech) and chemical compounds to test. At 2 or 3 days after culturing, T cells were stained with DAPI (Invitrogen) to remove the dead cells. We detected GFP and mRFP expression or GPR15 protein expression by flow cytometer (BD Fortessa). Full names of the compounds, their CAS numbers, the ranges of concentration tested, and their sources are listed in Table S2.

### Purification of human T cells and in vitro culture

Human naive T cells were isolated from anonymous donors' Leukoreduction (LR) filters. These filters were obtained from TJU's blood donation bank. The gender and age of the blood donors are unknown. LR filters were back-flushed with 60 mL buffer, and lymphocytes were collected in 50 mL conical tubes. Live cells were isolated via Ficoll-Paque density separation. CD14$^-$ cells were enriched through the MojoSort Human CD14 selection kit (Biolegend), and CD4$^+$ cells were enriched through MojoSort Human CD4 nanobeads (Biolegend). Subsequently, naive T cells (CD45RA$^+$CD25$^-$CD4$^+$CD3$^+$HLA-DR$^-$GPR15$^-$) were sorted by FACS (BD Aria II sorter) using the following antibody panel: FITC-CD45RA (Clone: HI100), PE-CD25 (Clone: BC96), PECy7-CD3ε (Clone:UCHT1), PerCpCy5- HLA-DR (Clone: LN3), APC-GPR15 (Clone: SA302A10), and APC780-CD4 (Clone: RPA-T4). Sorted cells were collected and cultured with Dynabeads Human T-activator CD3/CD28 beads at a 1:1 cell-to-bead ratio (ThermoFisher), varying concentration of AhR ligands, and with or without TGF-β (3 ng/ml) for 4–5 days in RPMI media described above. The following antibodies were used for the staining: APC-hGPR15 (Clone: SA302A10), PE-hFOXP3 (Clone: 236 A/E7, Thermofisher).

## Antibodies used

FITC-mCD19 (1:400 dilution; Clone: 1D3/CD19, Biolegend 152404), PerCP-Cy5.5-mTCRgd (1:400 dilution; Clone: GL3, Biolegend 118118), PE-mFOXP3 (1:100 dilution; Clone: FJK-16s, Thermofisher 12-5773-82), APC-mRORgt (1:100 dilution; Clone: B2D, Thermofisher 17-6981-82), Alexa 488-mHELIOS (1:100 dilution; Clone: 22F6, Biolegend 137223), Biotin-mGPR15 (1:20 dilution; Clone: S15042I, Biolegend 154602), Alexa 700-mI-A/I-E (1:800 dilution; Cone: M5/114.15.2, Biolegend 107622), APCFire-mTCRb (1:400 dilution; Clone: H57-597, Biolegend 109246), Pacific blue or BV421-mCD8b (1:1000 dilution; Clone: 53–5.8 or YTS156.7.7, Biolegend 140414 or 126629), BV605-mNK1.1 (1:400 dilution; Clone: PK136, Biolegend 108753), BV650-mCD45 (1:4000 dilution; Clone: 30-F11, Biolegend 103151), BV785-mCD4 (1:400 dilution; Clone: RM4-5, Biolegend 100552), mCD16/32 (1:200 dilution; Clone: 2.4G2, Tonbo Biosceinces 30-0161-U500), PE-mCD45.2 (1:400 dilution; Clone: 104, Biolegend 109808), Pacific-blue-mCD45.1 (1:400 dilution; Clone: A20, Biolegend 110722), APC-mTCRVb6 (1:400 dilution; Clone: RR4-7, Biolegend 140006), mCD3e (Clone: 145-2c11, Biolegend 100340), mCD28 (Clone: 37.51, Biolegend 102116), hamster IgG (polyclonal, MPbio 855398 or Sigma SAB3700488), PE-IFNg (1:100 dilution; Clone: XMG1.2, Biolegend 505808), APC-IL17A (1:100 dilution; Clone: eBio17B7, Thermofisher 17-7177-81), FITC-hCD45RA (1:33 dilution; Clone: HI100, Biolegend 304148), PE-hCD25 (1:20 dilution; Clone: BC96, Biolegend 302606), PECy7-hCD3e (1:33 dilution; Clone:UCHT1, Biolegend 300420), PerCPCy5.5-HLA-DR (1:33 dilution; Clone: LN3, Biolegend 327020), APC-hGPR15 (1:20 dilution; Clone: SA302A10, Biolegend or Clone 3679902, R&D), Alexa780-hCD4 (1:33 dilution; Clone: RPA-T4, Biolegend 300518), PE-hFOXP3 (1:100 dilution; Clone: 236 A/E7, Thermofisher 12-4777-42).

## Computer software

FACSDIVA software (Version 8, BD) was used to collect data from flow cytometers. Analysis of the flow cytometry data was done by FlowJo software (Version 10; Becton, Dickinson & Company). Graphic illustrations in Fig. 6f were created with BioRender.com.

## Statistical methods

Unless otherwise noted, statistical analysis was performed with unpaired or paired, two-tailed Student's t-test (for comparisons of two groups) by Prism software (Version 10; GraphPad). Results were shown as means±standard errors (SEM).

## Reporting summary

Further information on research design is available in the Nature Portfolio Reporting Summary linked to this article.

# Data availability

Source data generated in this study are provided in the Supplementary Information/Source Data file. The raw flow cytometry data generated in this study has been deposited in the flow repository database at http://flowrepository.org under accession codes as follows: FR-FCM-Z6S8 for Figs. 1b, c, and 3a (TJU1); FR-FCM-Z6S9 for Fig. 1d; FR-FCM-Z6SA for Figs. 1e, 3c (JAX), 6 f (mouse), and 6 g (mouse): FR-FCM-Z6SB for Fig. 2a–c; FR-FCM-Z6SC for Fig. 2e, f; FR-FCM-Z6S5 for Fig. 3b (TJU2); FR-FCM-Z6S4 for Fig. 3c (TAC); FR-FCM-Z6SD for Fig. 3d; FR-FCM-Z6SE for Fig. 3e; FR-FCM-Z6SF for Fig. 3f; FR-FCM-Z6T6, FR-FCM-Z6T7, FR-FCM-Z6T9, and FR-FCM-Z6TA for Fig. 4a; FR-FCM-Z6SL for Fig. 4b; FR-FCM-Z6TC for Fig. 5a; FR-FCM-Z6SX for Figs. 5c and S6; FR-FCM-Z6SP for Fig. 6f, g; FR-FCM-Z6SQ for Fig. S1; FR-FCM-Z6SR for Fig. S2; FR-FCM-Z6SS for Fig. S3; FR-FCM-Z6ST for Fig. S4a; FR-FCM-Z6SU for Fig. S4b; FR-FCM-Z6SV for Fig. S5; FR-FCM-Z6TE for Fig. S7a, c; FR-FCM-Z6TY for Fig. S7b. All other data are available in the article and its Supplementary files or from the corresponding author upon request. Source data are provided in this paper.

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

## Acknowledgements

We appreciate the technical support from the Sidney Kimmel Cancer Center (SKCC) Flow Cytometry Shared Resource, Translational Research/Pathology Shared Resource, and Laboratory Animal Shared Resource at Thomas Jefferson University. We thank the Department of Surgery, Division of Colorectal Surgery, and the Clinical Research Unit at Thomas Jefferson University for assistance in obtaining normal tissue samples from colorectal cancer patients. We thank D.R. Littman (New York University, New York) and M. Xu (National Institute of Biological Sciences, Beijing) for sharing HH7-2 TCR transgenic mice. We thank D.R. Littman and the Germ-free facility at New York University for sharing germ-free mice. We thank M.E. Fortini (SKCC, TJU) and L.V. Hooper (UT Southwestern) for reading our manuscript and providing feedback. We thank Pamela Walter (TJU) and Elizabeth Declan (TJU) for proofreading. This work was supported by NIAID grant R01AI141787 (S.V.K), NIAID grant R21AI142318 (S.V.K.), Career Development Award from Crohn's and Colitis Foundation of America (#329388) (S.V.K.), SKCC pilot fund for the iGONAD method of mouse genetic modification (L.J.S.), and National Institute of Health grant 5P30CA056036-22 (TJU SKCC core grant).

## Author contributions

S.V.K. conceptualized the study; designed and performed the experi-ments; analyzed the data; and co-wrote the manuscript with significant contributions from K.Z. and G.C.P. N.T.V. and K.Z. designed some experiments, performed most of the experiments, and analyzed the data. R.M.W. designed and performed some of the experiments. A.I.K. designed in vitro screening of AhR ligands, performed the experiments, and analyzed data. J.H.V. and G.J.O. performed independent validation experiments for in vitro screening results of AhR ligands. J.H. and M.A. performed histopathological analysis of the colitis. R.S.L, M.C., and A.E.S. provided data from human specimens and critical input to the manuscript. C.R.D., M.P., G.K., L.J.S., and G.C.P. generated mice used in this study and provided critical input to the manuscript.

## Competing interests
