## [Peer Review File · Nature Communications]

Dietary L-Tryptophan consumption determines the number of colonic regulatory T cells and susceptibility to colitis via GPR15REVIEWER COMMENTS

Reviewer #1 (Remarks to the Author):

This study from Van et al establishes a novel role for dietary L-Trp on colonic GPR15+ Treg generation. The study is very well designed and could lead to novel therapeutics aiming at increasing colonic Treg to alleviate inflammatory bowel diseases.

Addressing the following points would strengthen even more these findings:

1. Dietary metabolites usually regulate small intestine Treg, while gut microbiota-derived metabolites induce colonic Treg. Gut homing receptors CCR9 or α4β7 are induced in the mesenteric lymph nodes or Peyer's patches. What are the proportions/numbers of GPR15+ Treg in these sites as well as in the blood of mice supplemented with L-Trp vs controls?
2. A study by Yoshimatsu et al has shown that AhR activation in epithelial cells could induce colonic Helios + Treg and protect from colitis in mice (<https://doi.org/10.1016/j.celrep.2022.110773>). The transfer experiment in Fig 2 suggests that the Treg are peripherally induced by L-Trp however, it would be useful to identify whether L-Trp induces colonic Helios+ GPR15+ Treg (thymic derived) versus RORγt GPR15+ Treg (microbiota-derived) which will also confirm that the gut microbiota is not involved.
3. It would be useful to have non-infected control mice in Fig 2 to fully appreciate the role of L-Trp under challenging conditions.
4. While it takes 2 weeks for L-Trp to increase colonic GPR15+ Treg, it is unclear why 10 days instead of 2 weeks were chosen in Fig 2. It would be good to justify the timing of this experiment.
5. While L-trp increases colonic GPR15+ Treg it would be interesting to evaluate in these mice CCR9+ Treg and α4β7 Treg in the SI and colon along with GPR15+ Treg.
6. While the L-Trp supplementation in Abx-treated mice shows an effect independent of the

gut microbiota, it would be good to confirm this finding in germ-free mice. Moreover, what is the impact of L-Trp supplementation on the gut microbiota composition?

7. The identification of the AhR ligands involved in GPR15 induction is very interesting. While the impact of L-Trp seems independent of the gut microbiota according to Fig 2, microbes derived I3-PYA, I3S and pyocyanin induce GPR15+ Treg in vitro. Can the processing of L-Trp by the gut microbiota, during L-Trp supplementation, lead to the generation of these ligands? Are they naturally found in lower proportion in the colon of germ-free mice? The difficulty to identify the natural ligand for AhR in naïve T cells is understandable, is it possible to stimulate naïve T cells with L-trp and run a metabolomic analysis to identify which AhR ligands are generated?

8. L-Trp decreases the Th1 response, what about Th17? This is particularly relevant as Th17 is induced by *C. rodentium*. Showing the impact of Trp suppl on Th1 and Th17 in the colon of control vs infected mice would be a nice addition to the manuscript.

9. It is unclear how Fig 9b was generated. What is the n number of mice and humans used to compare their proportion of colonic GPR15+ Treg? How was the resulting plot done?

10. The therapeutic potential of L-Trp is discussed yet only a prophylactic approach was used in the study. What is the impact of L-trp supplementation when it is initiated the day after the *C. rodentium* infection. Regardless of the results the findings will be very interesting.

11. The benefits of L-Trp supplementation in colitis are clear but it would be valuable to discuss the potential side effects of increasing colonic Treg particularly in the context of colorectal cancer.

Reviewer #2 (Remarks to the Author):

In this manuscript, Van and colleagues investigated the effect of L-Tryptophan on GPR15+Treg generation in vivo via AhR signaling. The authors mainly used AhR conditional

KO mice and L-Tryptophan-supplemented/reduced diet to evaluate how GPR15+Treg cells developed. This is an interesting paper, which may contribute to better understanding of the function of AhR to control Treg generation and intestinal disorders such as ulcerative colitis associated with AhR deficiency. However, it has several essential problems which need to be addressed to improve the quality of the report.

Comments:

1. Regarding the AhR CD4-cKO condition, Yoshimatsu et al., Cell Reports 2022 (Fig.4) showed basically no significant changes in the frequency of total Treg in colonic LP. In this manuscript by Van et al, however, it seems that there was some change in total colonic Treg cells especially in Fig.5. The authors need to show not only GPR15+Foxp3+ or GPR15-Foxp3+ classification but also the frequency of colonic Treg cells throughout the manuscript.
2. Regarding AhR function for Treg cells, Treg specific AhR KO by utilizing the Foxp3-Cre system needs to be analyzed. The authors' experiments do not exclude the possibility of GPR15+Tconv conversion into GPR15+Treg cells in colonic LP, as a recent report has shown that some Th17 cells, called Treg-lineage committed Tconv cells, bear the potential to differentiate into Foxp3+Treg cells in a specific condition (van der Veecken et al., Immunity 2022).
3. In the colitis model (especially in Fig.6), the authors used infection-induced colitis with C.Rodentium. Is it suitable as an ulcerative colitis model?
4. In Fig.1a (and other figures), the authors indicate some change in the frequency of GPR15+Foxp3- fraction (Tconv cells) which was reduced in AhR cKO condition. In the whole manuscript, the author didn't analyze GPR15+ Tconv cells. Were they Th1, Th2, or Th17, for example? Do they produce Th specific cytokines ?
5. In Fig.4, FICZ and BaP differentially functioned as T cell polarizers; i.e., FICZ actively induced GPR15+Tconv cells while BaP induced GPR15+Treg cells. Could FICZ production occur in a specific condition in the presence of some microbiota and induce GPR15+Tconv cells in vivo? Also, does p.o. administration of BaP work as well as i.p. administration they performed?
6. In Fig.6d, the authors estimated the amount of L-tryptophan consumption in mice and humans and concluded that less GPR15+Treg cells in humans could be due to the difference of L-tryptophan consumption. However, just one representative data of FACS plot of GPR15

and FOXP3 expression is shown. How many samples were analyzed? This Fig.6d can be moved to Supplemental data.

7. In Fig.1e, the authors show that the L-Tryptophan-reduced food reduced GPR15+Treg cells in the spleen as well. This needs to be discussed.

8. In Fig.3, the authors show that GPR15+Foxp3⁻ fraction (Tconv) decreased in the GF condition whereas GPR15+Treg cells had no change. This means that GPR15+Treg cells developed in a microbiota-independent manner, or, conversely GPR15+Tconv cells development is microbiota-dependent. This needs to be discussed.

9. In Fig.5b, 3'distal Gpr15 locus contains not only AHR-binding but also AP-1-binding domains. Is AP-1 binding needed to express GPR15 in T cells including Treg and others?

Reviewer #3 (Remarks to the Author):

In the manuscript entitled Dietary L-Tryptophan consumption determines the number of colonic

regulatory T-cells and susceptibility to colitis via GPR15 by Van et al, the author investigated the role of L-Trp on colonic Foxp3⁺ Treg expression of GPR15. Several key findings are presented:

- The amount of dietary L-Trp correlated with colonic Foxp3⁺ Tregs numbers via the AhR-GPR15 axis
- L-Trp increased GPR15 during CD4⁺ T cell activation/migration to the LI via GPR15
- The ability of L-Trp to increase GPR15+Foxp3⁺ Tregs was dependent upon IDO1/2 but not microbiota
- AhR ligands have differential ability to induce Teff vs Tregs and AhR directly regulates GPR15
- L-Trp supplementation reduced Citrodentium colitis

While it is published that AhR regulates GPR15 expression and colonic Foxp3⁺ Tregs, it has been unclear which natural ligands may be regulating this pathway. The present study provides compelling evidence that L-Trp supplementation can regulate Foxp3⁺ Treg numbers in the colonic LP. The use of Ahr-CD4-CKO, Gpr15(gfp/gfp)Foxp3^{mrfp}, Cd4creAhrfl/fl, HH7-2TgRag1(n/n) and Ido1/2 DKO all provided supporting in vivo evidence for the role of the L-Trp pathway through IDO1/2 to induce GPR15. The experiments are all

very thorough and nicely presented.

One main concern I have is the choice of the model to test the ability of L-Trp to ameliorate colitis – the Citro rodentium model. This is a model of EPEC/EHEC (authors refer to this as enterotoxogenic?), but is not a good model for chronic intestinal inflammation that mimics human IBD (CD/UC). The data obtained using this model are only somewhat convincing. Several of the readout show no and minor difference between WT and GPR15 KO mice suggesting that GPR15+ endogenous Tregs are not playing a robust role in Citro infection. Overall, this is a poor model to test the hypothesis that L-Trp supplementation can ameliorate colitis. Other colitis models should be considered and explored. Similarly, only a prevention model was tested and a treatment model should be explored in order to demonstrate whether L-Trp supplementation can prevent and/or treat chronic colitis.

Jun 28th, 2023

Dear Reviewers:

On behalf of my colleagues, I would like to thank you all for your incredibly helpful suggestions to strengthen our findings. The following is a point-by-point response to your comments on our manuscript: **“Dietary L-tryptophan consumption determines the number of colonic regulatory T cells and susceptibility to colitis via GPR15.”** All changes to the manuscript were highlighted in yellow, either in the text or in the figure legends.

REVIEWER COMMENTS

Reviewer #1 (Remarks to the Author):

This study from Van et al establishes a novel role for dietary L-Trp on colonic GPR15+ Treg generation. The study is very well designed and could lead to novel therapeutics aiming at increasing colonic Treg to alleviate inflammatory bowel diseases.

Addressing the following points would strengthen even more these findings:

1. Dietary metabolites usually regulate small intestine Treg, while gut microbiota-derived metabolites induce colonic Treg. Gut homing receptors CCR9 of a4b7 are induced in the mesenteric lymph nodes or Peyer's patches. What are the proportions/numbers of GPR15+ Treg in these sites as well as in the blood of mice supplemented with L-Trp vs controls?

Thank you very much for your incredibly helpful suggestions to strengthen our manuscript. We have included new data in Supplementary Fig. 3b, demonstrating that mesenteric lymph nodes (MLNs) draining proximal and mid-colon (Houston et al. 2016, PMID: 26329428) contain increased GPR15+ Tregs in response to L-Trp supplementation. While the increase is small, it is likely because GPR15+ Tregs are constantly migrating out of the MLNs to the large intestine instead of staying in MLNs. We believe that the spleen CD4 T cell profile will mirror that of the blood, and there was no increase in the spleen (Fig. 1d).

2. A study by Yoshimatsu et al has shown that AhR activation in epithelial cells could induce colonic Helios + Treg and protect from colitis in mice (<https://doi.org/10.1016/j.celrep.2022.110773>). The transfer experiment in Fig 2 suggests that the Treg are peripherally induced by L-Trp however, it would be useful to identify whether L-Trp induces colonic Helios+ GPR15+ Treg (thymic derived) versus RORgt GPR15+ Treg (microbiota-derived) which will also confirm that the gut microbiota is not involved.

We have now included data indicating that the majority of colonic GPR15+ Tregs increasing in response to L-Trp supplementation at steady state are HELIOS+GPR15+ Tregs (Fig. 2c), which is consistent with our data indicating that host-derived L-Trp metabolites are sufficient to induce colonic GPR15+ Tregs (Fig. 3d-e). We also added a study by Yoshimatsu et al. 2022 (PMID: 35545035) as a reference for readers.

3. It would be useful to have non-infected control mice in Fig 2 to fully appreciate the role of L-Trp under challenging conditions.

Unfortunately, when transferred to a new host without *H. hepaticus*, HH7-2 TCR transgenic naive T cells do not survive for long, and on day 10 there were no donor cells left in the recipient mice.

4. While it takes 2 weeks for L-Trp to increase colonic GPR15+ Treg, it is unclear why 10 days instead of 2 weeks were chosen in Fig 2. It would be good to justify the timing of this experiment.

We tried to keep overall 13-14 days of elementary diet treatment before the analysis. To have L-Trp metabolite-enriched host environment at the time of T cell activation, we started elementary diet feeding 3 days before naïve donor cell transfer in Fig. 2d-f. To analyze the trafficking of donor T cells to the large intestine as soon as possible, we analyzed the mice at day 10 after the donor cell transfer. That makes up 13 days (close to 2 weeks overall). We added a short explanation for this in the revised version.

5. While L-tryptophan increases colonic GPR15⁺ Tregs it would be interesting to evaluate in these mice CCR9⁺ Treg and α4β7 Treg in the SI and colon along with GPR15⁺ Treg.

We have now included data (Figs. 2a-b, Supplementary Fig. 3) showing CCR9⁺ Tregs and GPR15⁺ Tregs in the small and the large intestine after L-tryptophan supplementation. As expected, L-tryptophan did not increase CCR9⁺ Tregs. α4β7 integrin is not the only dominant integrin for gut homing (Rivera-Nieves et al. 2005. PMID:15699171), and is involved in T cell homing to both SI and colon (Kim et al. 2013. PMID: 23661644). Therefore, we use CCR9 vs. GPR15 comparison instead.

6. While the L-tryptophan supplementation in Abx-treated mice shows an effect independent of the gut microbiota, it would be good to confirm this finding in germ-free mice. Moreover, what is the impact of L-tryptophan supplementation on the gut microbiota composition?

It has been shown that germ-free mice have poorly developed intestinal immune systems. Therefore, we reasoned that mice with normally developed intestinal immune system should be used to study the effect of L-tryptophan supplementation on the intestinal T cells. That was why Abx-treated mice were used in our experiments for L-tryptophan supplementation (Fig. 3e). Besides, elementary diets are not autoclavable (they will get burned), and therefore it is technically challenging to maintain germ-free isolators with unautoclaved elementary diets. Still, we used germ-free mice at steady state to confirm that the generation of colonic GPR15⁺ Tregs is independent of gut microbiota (Fig. 3d). Furthermore, IDO1/2 double KO mice showed decreased colonic GPR15⁺ Tregs, confirming that host-derived L-tryptophan metabolites are the key factors for the generation of GPR15⁺ Tregs (Fig. 3f).

Regarding gut microbial composition, we initially performed metagenomic sequencing after elementary diet treatment, and we found that 2-week treatment of elementary diets in our mouse facility did cause some changes in microbial gene expression. However, since our phenotype is independent of microbiota, we did not pursue it further in this study. We plan to study the impact of L-tryptophan metabolites from gut microbiota in the follow-up study, since FOXP3-GPR15⁺CD4⁺ T cells seem to require microbiota either as a source of TCR antigens or L-tryptophan metabolites (Fig. 3d). But we believe that it is beyond the scope of this study.

7. The identification of the AhR ligands involved in GPR15 induction is very interesting. While the impact of L-tryptophan seems independent of the gut microbiota according to Fig 2, microbes derived I3-PYA, I3S and pyocyanin induce GPR15⁺ Treg in vitro. Can the processing of L-tryptophan by the gut microbiota, during L-tryptophan supplementation, lead to the generation of these ligands? Are they naturally found in lower proportion in the colon of germ-free mice? The difficulty to identify the natural ligand for AhR in naïve T cells is understandable, is it possible to stimulate naïve T cells with L-tryptophan and run a metabolomic analysis to identify which AhR ligands are generated?

L-tryptophan may be converted into I3-PYA, I3S, and pyocyanin if the gut microbes can produce these metabolites (I3S still needs host-mediated conversion of microbe-derived indoles). Therefore, it is certainly possible that I3-PYA, I3S, and pyocyanin play a role in GPR15 expression in vivo. However, it is extremely unlikely that they are the key factors in the development of GPR15⁺ Tregs in our experimental settings due to the following reasons: First, it is very well established that 95% of L-tryptophan ingested by the host is metabolized through the IDO1/2 pathway, and only 4-6% is used by microbiota (Gao et al. 2018. PMID: 29468141). Second, as depicted in Fig. 3, GPR15⁺ Treg production and response to L-tryptophan were perfectly normal in the absence of microbiota (Fig. 3d-e), and host-derived L-tryptophan metabolites via the IDO1/2 pathway are necessary for the generation GPR15⁺ Tregs (Fig. 3f). Third, there is also no evidence indicating that I3-PYA, I3S, and pyocyanin can be converted from host IDO1/2 metabolites as far as we know.

Regarding L-tryptophan in vitro, L-tryptophan itself did not stimulate GPR15 expression in T cell cultures in vitro (results were omitted from Fig. 4 because L-tryptophan is not considered as a potential AhR ligand), demonstrating that L-tryptophan metabolites are produced by cells other than T cells. We believe that identifying the cells that produce relevant L-tryptophan metabolites and conducting a metabolomic analysis are beyond the scope of this study.

8. L-tryptophan decreases the Th1 response, what about Th17? This is particularly relevant as Th17 is induced by *C. rodentium*. Showing the impact of Trp suppl on Th1 and Th17 in the colon of control vs infected mice would be a nice addition to the manuscript.

Segmented filamentous bacteria (SFB) affects the number of Th17 cells in the intestinal lamina propria at steady state. Mice maintained in our colonies were all SFB-negative and therefore it was not easy to detect a meaningful level of Th17 cells at steady state (Supplementary Fig. 7b). We have now added new data showing IL-17A producing cells in the large intestine of Taconic C57BL/6 mice after L-tryptophan supplementation (Supplementary Fig. 7c). While the level of CD4⁺IL17A⁺ T cells did not change with Trp-Sup, the level of IFNγ⁺IL17A⁺CD4⁺ T cells, IL17A⁺ gdT cells as well as IFNγ⁺CD4⁺ T cells were significantly reduced upon Trp-Sup dietary treatment. Unfortunately, SFB-containing C57BL/6 mice (such as ones from Taconic or Charles River) are resistant to *C. rodentium*-induced colitis due to the extensive presence of Th17 cells in the intestine. Therefore, we could not use them in *C. rodentium*-induced colitis model.

9. It is unclear how Fig 9b was generated. What is the n number of mice and humans used to compare their proportion of colonic GPR15⁺ Treg? How was the resulting plot done?

We have now added graphs representing each population of CD4⁺ T cells based on GPR15 and FOXP3 expression (Fig. 6g). We have used 5 sample from mice and 6 samples from humans as stated in the figure legend.

10. *The therapeutic potential of L-Trp is discussed yet only a prophylactic approach was used in the study. What is the impact of L-trp supplementation when it is initiated the day after the C rodentium infection. Regardless of the results the findings will be very interesting.*

We have now added new data showing the impact of L-Trp supplementation after the onset of colitis (Supplementary Fig. 8). Day 5 after *C.rodentium* infection was when mice started to have diarrhea and chosen as a start date for L-Trp treatment. Unfortunately, L-Trp supplementation after the onset of colitis did not help to alleviate the symptoms (Supplementary Fig. 8).

11. *The benefits of L-Trp supplementation in colitis are clear but it would be valuable to discuss the potential side effects of increasing colonic Treg particularly in the context of colorectal cancer.*

GPR15⁺ Tregs were detected colorectal cancer lesions in both mouse and human (Adamczyk et al. 2021. PMID 33727229). Therefore, L-Trp supplementation to increase GPR15⁺ colonic Tregs may not be helpful for the prognosis of colorectal cancer. We added comments in the discussion as the reviewer suggested.

Reviewer #2 (Remarks to the Author):

In this manuscript, Van and colleagues investigated the effect of L-Tryptophan on GPR15+Treg generation in vivo via AhR signaling. The authors mainly used AhR conditional KO mice and L-Tryptophan-supplemented/reduced diet to evaluate how GPR15+Treg cells developed. This is an interesting paper, which may contribute to better understanding of the function of AhR to control Treg generation and intestinal disorders such as ulcerative colitis associated with AhR deficiency. However, it has several essential problems which need to be addressed to improve the quality of the report.

Comments:

1. *Regarding the AhR CD4-cKO condition, Yoshimatsu et al., Cell Reports 2022 (Fig.4) showed basically no significant changes in the frequency of total Treg in colonic LP. In this manuscript by Van et al, however, it seems that there was some change in total colonic Treg cells especially in Fig.5. The authors need to show not only GPR15+Foxp3+ or GPR15-Foxp3+ classification but also the frequency of colonic Treg cells throughout the manuscript.*

Thank you very much for your incredibly helpful suggestions to strengthen our manuscript. As the reviewer suggested, we have included the plots for total colonic Treg frequencies throughout the manuscript. There are many factors that may have affected the results in the paper that the reviewer mentioned (Yoshimatsu et al. 2022. PMID: 35545035), such as the differences between CD4-CreERT2 vs. CD4-Cre, and the potential toxicity of >150 mg/kg of Tamoxifen in GI tract. However, we think it is more likely that the tryptophan level in the diet determined the chance of detecting the effect of AhR deficiency in Tregs. For example, it is clear that the dependency of colonic Tregs on AhR is more visible in the percentage of cells when TrpSup diet was provided to mice (Fig. 1c). We also added a study by Yoshimatsu et al. 2022 (PMID: 35545035) as a reference for readers.

2. *Regarding AhR function for Treg cells, Treg specific AhR KO by utilizing the Foxp3-Cre system needs to be analyzed. The authors' experiments do not exclude the possibility of GPR15+Tconv conversion into GPR15+Treg cells in colonic LP, as a recent report has shown that some Th17 cells, called Treg-lineage committed Tconv cells, bear the potential to differentiate into Foxp3+Treg cells in a specific condition (van der Veeke et al., Immunity 2022).*

We have now included data in Fig. 2c showing that the increase of GPR15⁺ Tregs upon L-Trp supplementation at steady state mainly occurs through the increase of thymus-derived Tregs (HELIOS⁺). In addition, we used SFB-free mice in most of our experiments, which have very few Th17 cells in the intestine. Therefore, it is unlikely that the conversion of Treg-lineage committed Tconv Th17 to GPR15⁺ Tregs contributes significantly to the phenotype in our experimental conditions. However, it is still possible that such conversion may occur in other cases. Therefore, we mentioned that in the discussion and cited the paper that the reviewer mentioned.

3. *In the colitis model (especially in Fig.6), the authors used infection-induced colitis with C.Rodentium. Is it suitable as an ulcerative colitis model?*

We have chosen to use *C. rodentium* model since its colitis severity inversely correlates with the number of colonic Tregs (Kim et al. 2013. PMID: 23661644), the effect of which was what we need to test in our L-Trp supplementation experiments. However, it is also known that none of the single colitis models in mice reflect all the features of human IBD (Katsandegwaza et al. 2022. PMID: 36012618). Therefore, in response to the reviewer's comment, we have added data from the most widely used colitis model (DSS-induced colitis). We found that 2-week L-Trp supplementation can alleviate the symptoms of DSS-induced colitis (Figure 6d-e), consistent with our results from *C. rodentium* model (Figure 6b-c).

4. In Fig. 1a (and other figures), the authors indicate some change in the frequency of GPR15⁺Foxp3⁻ fraction (Tconv cells) which was reduced in AhR cKO condition. In the whole manuscript, the author didn't analyze GPR15⁺ Tconv cells. Were they **Th1, Th2, or Th17**, for example? Do they produce Th specific cytokines?

We have included new data in Supplementary Fig. 7a, showing that GPR15⁺FOXP3⁻ cells do express IFN γ or IL-17A (but in our mouse colony, we could not detect IL-4⁺ cells in the large intestine at steady state). However, most IFN γ or IL-17A producing T helper cells were GPR15⁻ (Supplementary Fig. 7a).

5. In Fig. 4, FICZ and BaP differentially functioned as T cell polarizers; i.e., FICZ actively induced GPR15⁺Tconv cells while BaP induced GPR15⁺Treg cells. Could **FICZ production** occur in a specific condition in the presence of some microbiota and induce GPR15⁺Tconv cells in vivo? Also, does p.o. administration of BaP work as well as i.p. administration they performed?

FICZ production by microbiota is a poorly researched topic, thus the authors are unable to say if FICZ production can occur in the presence of some microbiota. Therefore, we left it as an open possibility in the revised discussion section. We did not evaluate the effect of BaP administration by gavage because BaP can affect microbiota (He et al., 2019, PMID 31732054) and therefore can complicate the interpretation of the results. We believed studying the in vivo effect of direct exposure to BaP was a priority. We believe that the effect of BaP ingestion can be studied in a separate research project in combination with its impact on microbiota.

6. In Fig. 6d, the authors estimated the amount of L-tryptophan consumption in mice and humans and concluded that less GPR15⁺Treg cells in humans could be due to the difference of L-tryptophan consumption. However, just one representative data of FACS plot of GPR15⁺ and FOXP3 expression is shown. **How many samples were analyzed?** This Fig. 6d can be moved to Supplemental data.

We have included graphs showing each data point for GPR15 and FOXP3 expression among CD4⁺ T cells (Figure 6g). In total, 5 (mouse) and 6 (human) samples were used.

7. In Fig. 1e, the authors show that the L-Tryptophan-reduced food reduced GPR15⁺Treg cells in the **spleen** as well. This needs to be discussed.

We have included a discussion in the revised manuscript. Trp-def diet reduced both GPR15⁺ and GPR15⁻ Tregs in the spleen regardless of AhR. We think in the absence of L-Trp, AhR- and GPR15-independent phenotype predominates in the spleen.

8. In Fig. 3, the authors show that GPR15⁺Foxp3⁻ fraction (Tconv) decreased in the GF condition whereas GPR15⁺Treg cells had no change. This means that **GPR15⁺Treg cells developed in a microbiota-independent manner, or, conversely GPR15⁺Tconv cells development is microbiota-dependent.** This needs to be discussed.

That is certainly an important finding to be discussed. We have added that in the revised manuscript.

9. In Fig. 5b, 3' distal Gpr15 locus contains not only AHR-binding but also AP-1-binding domains. Is **AP-1 binding** needed to express GPR15 in T cells including Treg and others?

TCR stimulation is necessary to observe the induction of GPR15 expression in vitro in our experiments (Fig. 4a). Therefore, it is likely that AP-1 plays a role in some parts. However, it is not clear whether the potential AP-1 binding sites near AHR binding motifs are needed for GPR15 induction. We believe finding that out is beyond the scope of this study.

Reviewer #3 (Remarks to the Author):

In the manuscript entitled Dietary L-Tryptophan consumption determines the number of colonic regulatory T-cells and susceptibility to colitis via GPR15 by Van et al, the author investigated the role of L-Trp on colonic Foxp3⁺ Treg expression of GPR15. Several key findings are presented:

- The amount of dietary L-Trp correlated with colonic Foxp3⁺ Tregs numbers via the AhR-GPR15 axis
- L-Trp increased GPR15 during CD4⁺ T cell activation/migration to the LI via GPR15
- The ability of L-Trp to increase GPR15⁺Foxp3⁺ Tregs was dependent upon IDO1/2 but not microbiota
- AhR ligands have differential ability to induce Teff vs Tregs and AhR directly regulates GPR15
- L-Trp supplementation reduced *Citro rodentium* colitis

While it is published that AhR regulates GPR15 expression and colonic Foxp3⁺ Tregs, it has been unclear which natural ligands may be regulating this pathway. The present study provides compelling evidence that L-Trp supplementation can regulate Foxp3⁺ Treg numbers in the colonic LP. The use of Ahr-CD4-CKO, Gpr15(gfp/gfp)Foxp3^{mrfp}, Cd4creAhr^{fl/fl}, HH7-2TgRag1(n/n) and Ido1/2 DKO all provided supporting in vivo evidence for the role of the L-Trp pathway through IDO1/2 to induce GPR15. The experiments are all very thorough and nicely presented.

*One main concern I have is the choice of the model to test the ability of L-Trp to ameliorate colitis – the *Citro rodentium* model. This is a*

model of EPEC/EHEC (authors refer to this as enterotoxogenic?), but is not a good model for chronic intestinal inflammation that mimics human IBD (CD/UC). The data obtained using this model are only somewhat convincing. Several of the readout show no and minor difference between WT and GPR15 KO mice suggesting that GPR15+ endogenous Tregs are not playing a robust role in Citro infection. Overall, this is a poor model to test the hypothesis that L-Trp supplementation can ameliorate colitis. Other colitis models should be considered and explored.

Thank you very much for your incredibly helpful suggestions to strengthen our manuscript. We have chosen to use *C. rodentium* model since its colitis severity inversely correlates with the number of colonic Tregs (Kim et al. 2013. PMID: 23661644), the effect of which was what we need to test in our experiments. However, it is also well known that none of the single colitis models in mice reflect all the features of human IBD (Katsandegwaza et al. 2022. PMID: 36012618). In that sense, the reviewer's suggestion to test our hypothesis in a different model of colitis was valuable. Therefore, we have added data from the most widely used colitis model (DSS-induced colitis) in Fig. 6. We found that 2-week L-Trp supplementation can alleviate the symptoms of DSS-induced colitis (Fig. 6d-e), consistent with our results from *C. rodentium* model (Figure 6b-c).

*Similarly, only a prevention model was tested and a **treatment model** should be explored in order to demonstrate whether L-Trp supplementation can prevent and/or treat chronic colitis.*

We now have included new data with a treatment model as the reviewer suggested. We started L-Trp supplementation at day 5 after the *C. rodentium* infection, when the colitis symptoms started to show up, and found that L-Trp supplementation after colitis onset did not alleviate the symptoms (Supplementary Fig. 8)

We hope this revision satisfies the editors and the reviewers, so that we can share our exciting discovery about the connection between dietary L-Tryptophan and colonic T cell responses with general public.

REVIEWER COMMENTS

Reviewer #1 (Remarks to the Author):

All my concern were addressed.

Reviewer #2 (Remarks to the Author):

In this revision, Van and colleagues have adequately addressed many concerns of this reviewer. However, there are still some small concerns still to be addressed.

Comments:

1. In Supplemental Fig. 3a, L-Trp supplementation didn't affect the frequency of GPR15+ Treg cells in SILP, meaning that L-Trp specifically functioned upon the production of GPR15+ Treg cells in LILP. The authors mentioned this phenomenon and similar results, "Only WT GPR15+FOXP3+ donor cells increased upon L-Trp supplementation, and this phenotype was found specifically in the LILP but not in the SILP...", but should discuss why L-Trp differentially affects GPR15+ Treg production in the different intestinal tracts.
2. According to Fig. 2c, the authors mention "We also found that most of the colonic GPR15+ Tregs that increased after L-Trp supplementation were thymus-derived HELIOS+ Tregs, as opposed to peripherally-derived RORgt+ Tregs". Helios was defined as one of thymic Treg markers in the paper by Thornton et al., 2010; however, it was also expressed on peripherally derived Treg cells, and several studies dealt with Helios as an activation-related marker in vivo Treg cells (Gottschalk et al., 2014). In this manuscript, the authors didn't address whether GPR15+ Tregs are truly thymus-derived or not. Their findings should be more careful in this point.
3. Related to the comment above, the authors didn't show a representative FACS plot of Helios and RORgt expression on GPR15+ Treg cells in Fig. 2c. Several studies indicated that colon Treg cells possess reciprocal expression of these markers in different physiological conditions (Ohnmacht et al., 2015; Pratama et al., 2019). The authors need to address this point by showing a relevant FACS figure.
4. In Fig. 6g, human GPR15+Foxp3- T cells were apparently enriched in colonic lamina propria compared to mouse colon, but the frequency of these cells highly varied. This

suggests that the variation of GPR15+ Tconv cells might reflect the diversity of gut microbiota. This topic is also related to Fig. 3 and my previous comment #8. The authors may need to discuss this.

References

Thornton et al., J Immunol 2010 (DOI: 10.4049/jimmunol.0904028)

Gottschalk et al., J Immunol 2014 (DOI: 10.4049/jimmunol.1102964)

Ohnmacht et al., Science 2015 (DOI: 10.1126/science.aac4263)

Pratama et al., J Exp Med 2019 (DOI: 10.1084/jem.20190428)

Reviewer #3 (Remarks to the Author):

The authors have thoughtfully and thoroughly responded to the critiques. Given that L-Trp supplementation failed to treat murine colitis, this point should be acknowledged in the discussion as a potential limitation to using L-Trp for treating human IBD.

Aug 10th, 2023

Dear Reviewers:

On behalf of my colleagues, I would like to thank you all for your incredibly helpful suggestions to improve our manuscript. The following is a point-by-point response to your comments on our manuscript: **“Dietary L-tryptophan consumption determines the number of colonic regulatory T cells and susceptibility to colitis via GPR15.”** All changes to the manuscript were highlighted in yellow, either in the text or in the figure legends.

REVIEWER COMMENTS

Reviewer #2 (Remarks to the Author):

In this revision, Van and colleagues have adequately addressed many concerns of this reviewer. However, there are still some small concerns still to be addressed.

Comments:

1. In Supplemental Fig. 3a, L-Trp supplementation didn't affect the frequency of GPR15+ Treg cells in SILP, meaning that L-Trp specifically functioned upon the production of GPR15+ Treg cells in LILP. The authors mentioned this phenomenon and similar results, “Only WT GPR15+FOXP3+ donor cells increased upon L-Trp supplementation, and this phenotype was found specifically in the LILP but not in the SILP...”, but should discuss why L-Trp differentially affects GPR15+ Treg production in the different intestinal tracts.

Thank you very much for the helpful suggestions to improve our manuscript. GPR15 functions as a homing receptor specific for the large intestine, but not for the small intestine (Kim et al. 2013. PMID: 23661644). Therefore, L-Trp can differentially influence Treg population in the different intestinal tracts via the induction of GPR15 and subsequent action of GPR15 in T cell homing. As shown in Supplementary Fig. 2, L-Trp-mediated increase of GPR15+ Tregs in the large intestine only occurs in the presence of GPR15. While this point was explained in the text relevant to Supplementary Fig. 2, we concur with the reviewer that it should be emphasized. In the revised version, we modified our discussion to emphasize this point.

2. According to Fig. 2c, the authors mention “We also found that most of the colonic GPR15+ Tregs that increased after L-Trp supplementation were thymus-derived HELIOS+ Tregs, as opposed to peripherally-derived RORgt+ Tregs”. Helios was defined as one of thymic Treg markers in the paper by Thornton et al., 2010; however, it was also expressed on peripherally derived Treg cells, and several studies dealt with Helios as an activation-related marker in vivo Treg cells (Gottschalk et al., 2014). In this manuscript, the authors didn't address whether GPR15+ Tregs are truly thymus-derived or not. Their findings should be more careful in this point.

We agreed with the reviewer and modified our text to avoid misinterpretation of our data. In addition, we described a potential alternative origin of HELIOS+ Tregs and added a reference for readers.

3. Related to the comment above, the authors didn't show a representative FACS plot of Helios and RORgt expression on GPR15+ Treg cells in Fig. 2c. Several studies indicated that colon Treg cells possess reciprocal expression of these markers in different physiological conditions (Ohnmacht et al., 2015; Pratama et al., 2019). The authors need to address this point by showing a relevant FACS figure.

As the reviewer suggested, we have now included representative FACS plots in Fig. 2c.

4. In Fig. 6g, human GPR15+Foxp3- T cells were apparently enriched in colonic lamina propria compared to mouse colon, but the frequency of these cells highly varied. This suggests that the variation of GPR15+ Tconv cells might reflect the diversity of gut microbiota. This topic is also related to Fig. 3 and my previous comment #8. The authors may need to discuss this.

As the reviewer suggested, we have now included a discussion about human GPR15+FOXP3- T cells and the likely impact of diverse microbiota.

References

Thornton et al., *J Immunol* 2010 (DOI: 10.4049/jimmunol.0904028)
Gottschalk et al., *J Immunol* 2014 (DOI: 10.4049/jimmunol.1102964)
Ohnmacht et al., *Science* 2015 (DOI: 10.1126/science.aac4263)
Pratama et al., *J Exp Med* 2019 (DOI: 10.1084/jem.20190428)

Reviewer #3 (Remarks to the Author):

The authors have thoughtfully and thoroughly responded to the critiques. Given that L-Trp supplementation failed to treat murine colitis, this point should be acknowledged in the discussion as a potential limitation to using L-Trp for treating human IBD.

Thank you very much for the helpful suggestions to improve our manuscript. As the reviewer suggested, we have now included a discussion about a potential limitation to using L-Trp for treating human IBD.

We hope this revision satisfies the editors and the reviewers, so that we can share our exciting discovery about the connection between dietary L-Tryptophan and colonic T cell responses with general public.

REVIEWERS' COMMENTS

Reviewer #2 (Remarks to the Author):

They have properly responded to my comments.

Reviewer #3 (Remarks to the Author):

The authors have satisfactorily addressed my remaining concern.